# An unusual and vital protein with guanylate cyclase and P4-ATPase domains in a pathogenic protist

Özlem Günay-Esiyok, Ulrike Scheib, Matthias Noll, Nishith Gupta

cGMP signaling is one of the master regulators of diverse functions in eukaryotes; however, its architecture and functioning in protozoans remain poorly understood. Herein, we report an exclusive guanylate cyclase coupled with N-terminal P4-ATPase in a common parasitic protist, *Toxoplasma gondii*. This bulky protein (477-kD), termed *Tg*ATPase_P-GC to fairly reflect its envisaged multifunctionality, localizes in the plasma membrane at the apical pole of the parasite, whereas the corresponding cGMP-dependent protein kinase (*Tg*PKG) is distributed in the cytomembranes. *Tg*ATPase_P-GC is refractory to genetic deletion, and its CRISPR/Cas9–assisted disruption aborts the lytic cycle of *T. gondii*. Besides, Cre/loxP–mediated knockdown of *Tg*ATPase_P-GC reduced the synthesis of cGMP and inhibited the parasite growth due to impairments in the motility-dependent egress and invasion events. Equally, repression of *Tg*PKG by a similar strategy recapitulated phenotypes of the *Tg*ATPase_P-GC–depleted mutant. Notably, despite a temporally restricted function, *Tg*ATPase_P-GC is expressed constitutively throughout the lytic cycle, entailing a post-translational regulation of cGMP signaling. Not least, the occurrence of *Tg*ATPase_P-GC orthologs in several other alveolates implies a divergent functional repurposing of cGMP signaling in protozoans, and offers an excellent drug target against the parasitic protists.

## Introduction

cGMP is regarded as a common intracellular second messenger, which relays endogenous and exogenous cues to the downstream mediators (kinases, ion channels, etc.), and thereby regulates a range of cellular processes in prokaryotic and eukaryotic organisms (Lucas et al, 2000; Hall & Lee, 2018). The synthesis of cGMP from GTP is catalyzed by a guanylate cyclase (GC). Levels of cGMP are strictly counterbalanced by phosphodiesterase enzyme (PDE), which degrades cGMP into GMP by hydrolyzing the 3'-phosphoester bond (Beavo, 1995). PKG (or cGMP-dependent protein kinase) on the other hand is a major mediator of cGMP signaling in most eukaryotic cells; it phosphorylates a repertoire of effector proteins to exert a

consequent subcellular response. All known PKGs belong to the serine/threonine kinase family (Lucas et al, 2000). Much of our understanding of cGMP-induced transduction is derived from higher organisms, namely, mammalian cells, which harbor four soluble GC subunits ($\alpha_1$, $\alpha_2$, $\beta_1$, and $\beta_2$) functioning as heterodimers, and seven membrane-bound GCs (GC-A to GC-G), occurring mostly as homodimers (Lucas et al, 2000; Potter, 2011). There are two variants of PKG (PKG I and PKG II) reported in mammals. The type I PKGs have two soluble alternatively spliced isoforms ($\alpha$ and $\beta$) functioning as homodimers, whereas the type II PKGs are membrane-bound proteins (MacFarland, 1995; Pilz & Casteel, 2003), which form apparent monomers (De Jonge, 1981) as well as dimers (Vaandrager et al, 1997).

The cGMP pathway in protozoans shows a marked divergence from mammalian cells (Linder et al, 1999; Gould & de Koning, 2011; Hopp et al, 2012). One of the protozoan phylum Apicomplexa comprising >6,000 endoparasitic, mostly intracellular, species of significant clinical importance (Adl et al, 2007) exhibits even more intriguing design of cGMP signaling. *Toxoplasma*, *Plasmodium*, and *Eimeria* are some of the key apicomplexan parasites causing devastating diseases in humans and animals. These pathogens display a complex lifecycle in nature assuring their successful infection, reproduction, stage-conversion, adaptive persistence, and interhost transmission. cGMP cascade has been shown as one of the most central mechanisms to coordinate the key steps during the parasitic lifecycle (Gould & de Koning, 2011; Govindasamy et al, 2016; Baker et al, 2017; Brown et al, 2017; Frénal et al, 2017). In particular, the motile parasitic stages, for example, sporozoite, merozoite, ookinete and tachyzoite deploy cGMP signaling to enter or exit host cells (Baker et al, 2017; Frénal et al, 2017) or traverse tissues by activating secretion of micronemes (apicomplexan-specific secretory organelle) (Brochet et al, 2014; Brown et al, 2016; Bullen et al, 2016). Micronemes secrete adhesive proteins required for the parasite motility and subsequent invasion and egress events (Brochet et al, 2014; Brown et al, 2016; Bullen et al, 2016; Frénal et al, 2017), which are regulated by PKG activity.

The work of Gurnett et al (2002) demonstrated that *Toxoplasma gondii* and *Eimeria tenella* harbor a single PKG gene encoding for two alternatively translated isoforms (soluble and membrane-bound). The physiological essentiality of PKG for the asexual reproduction of both parasites was first revealed by a chemical-genetic

Institute of Biology, Faculty of Life Sciences, Humboldt University, Berlin, Germany

Correspondence: Gupta.Nishith@hu-berlin.de

approach (Donald et al, 2002), whereas the functional importance of this protein for secretion of micronemes, motility, and invasion of *T. gondii* tachyzoites and *E. tenella* sporozoites was proven by Wiersma et al (2004). Successive works in *T. gondii* have endorsed a critical requirement of *Tg*PKG for its asexual reproduction by various methods (Donald et al, 2002; Lourido et al, 2012; Sidik et al, 2014; Brown et al, 2017). Evenly, PKG is also needed for the hepatic and erythrocytic development of *Plasmodium* species (Falae et al, 2010; Taylor et al, 2010; Baker et al, 2017). It was shown that PKG triggers the release of calcium from the storage organelles in *Plasmodium* (Singh et al, 2010) and *Toxoplasma* (Brown et al, 2016). Calcium can in turn activate calcium-dependent protein kinases and exocytosis of micronemes (Billker et al, 2009; Lourido et al, 2012). The effect of cGMP signaling on calcium depends on inositol 1,4,5-triphosphate ($IP_3$), which is produced by phosphoinositide-phospholipase C, a downstream mediator of PKG (Brochet et al, 2014). Besides $IP_3$, DAG is generated as a product of phosphoinositide-phospholipase C and converted to phosphatidic acid, which can also induce microneme secretion (Bullen et al, 2016). On the other hand, cAMP-dependent protein kinase acts as a repressor of PKG and $Ca^{2+}$ signaling, thereby preventing microneme secretion as well as a premature egress (Jia et al, 2017; Uboldi et al, 2018).

Unlike the downstream signaling events, the onset of cGMP cascade remains underappreciated in Apicomplexa, partly because of a complex structure of GCs, as described in *Plasmodium* (Linder et al, 1999; Baker, 2004). Two distinct GCs, *Pf*GCα and *Pf*GCβ (Carucci et al, 2000; Baker, 2004; Hopp et al, 2012) were identified in *Plasmodium falciparum*. Lately, Gao et al (2018) demonstrated the essential role of GCβ for the motility and transmission of *Plasmodium yoelii*. Herein, we aimed to characterize an unusual GC fused with a P4-ATPase domain in *T. gondii,* and test the physiological importance of cGMP signaling for asexual reproduction of its acutely infectious tachyzoite stage. Our findings along with other independent studies published just recently (Brown & Sibley, 2018; Bisio et al, 2019; Yang et al, 2019), as discussed elsewhere, provide significant new insights into cGMP signaling during the lytic cycle of *T. gondii*.

## Results

### *T. gondii* encodes an alveolate-specific GC linked to P-type ATPase

Our genome searches identified a single putative GC in the parasite database (ToxoDB) (Gajria et al, 2008), comprising multiple P-type ATPase motifs at its N terminus and two nucleotide cyclase domains (termed as GC1 and GC2 based on the evidence herein) at the C terminus. Given the predicted multifunctionality of this protein, we named it *Tg*ATPase$_P$-GC. The entire gene size is about 38.3 kb, consisting of 53 introns and 54 exons. The ORF encodes for a remarkably large protein (4,367 aa, 477 kD), comprising P-type ATPase (270 kD) and nucleotide cyclase (207-kD) domains and includes 22 transmembrane helices (TMHs) (Fig 1A). The first half of *Tg*ATPase$_P$-GC (1–2,480 aa) contains 10 α-helices and four conserved ATPase-like subdomains: (i) the region from Lys$^{110}$ to His$^{174}$ encodes a

potential lipid-translocating ATPase; (ii) the residues from Leu$^{207}$ to Gly$^{496}$ are predicted to form a bifunctional E1-E2 ATPase binding to both metal ions and ATP, and thus functioning like a cation-ATPase; (iii) the amino acids from Thr$^{1647}$ to Ser$^{1748}$ harbor yet another metal-cation transporter with an ATP-binding region; (iv) the region from Cys$^{2029}$ to Asn$^{2480}$ contains a haloacid dehalogenase-like hydrolase, or otherwise a second lipid-translocating ATPase (Fig 1A).

The second half (2,481–4,367 aa) encodes a putative GC comprising GC1 and GC2 domains from Ser$^{2942}$-Lys$^{3150}$ and Thr$^{4024}$-Glu$^{4159}$ residues, respectively (Fig 1A). Both GC1 and GC2 follow a transmembrane region, each with six helices. The question-marked helix (2,620–2,638 aa) antecedent to GC1 has a low probability (score, 752). An exclusion of this helix from the envisaged model, however, results in a reversal of GC1 and GC2 topology (facing outside the parasite), which is unlikely given the intracellular transduction of cGMP signaling via *Tg*PKG. Moreover, our experiments suggest that the C terminus of *Tg*ATPase$_P$-GC faces inwards (see Fig 2B). Phylogenetic study indicated an evident clading of *Tg*ATPase$_P$-GC with homologs from parasitic (*Hammondia*, *Eimeria*, and *Plasmodium*) and free-living (*Tetrahymena*, *Paramecium*, and *Oxytricha*) alveolates (Fig S1). In contrast, GCs from the metazoan organisms (soluble and receptor-type) and plants formed their own distinct clusters. Quite intriguingly, the protist clade contained two groups, one each for apicomplexans and ciliates, implying a phylum-specific evolution of *Tg*ATPase$_P$-GC orthologs.

### P-type ATPase domain of *Tg*ATPase$_P$-GC resembles P4-ATPases

The N terminus of *Tg*ATPase$_P$-GC covering 10 TMHs and four conserved motifs is comparable with P4-ATPases, a subfamily of P-type ATPases involved in translocation of phospholipids across the membrane bilayer and vesicle trafficking in the secretory pathways (Palmgren & Nissen, 2011; Andersen et al, 2016). The human genome contains 14 different genes for P4-ATPases clustered in five classes (1a, 1b, 5, 2, and 6), all of which have five functionally distinct domains (Andersen et al, 2016): A (actuator), N (nucleotide binding), and P (phosphorylation) domains are cytoplasmic; whereas T (transport) and S (class-specific support) domains are membrane-anchored. Besides, a regulatory (R-) domain usually exists either at the N- or C terminus or at both ends (Palmgren & Nissen, 2011). In mammalian orthologs, the region between TMH1–TMH6 constitutes a functional unit for lipid flipping, and the segment between TMH7 and TMH10 undertakes a supportive role. ATPases have an intrinsic kinase activity to phosphorylate itself at the aspartate residue located in the P domain while catalytic cycle is taking place and later get dephosphorylated by the A domain when transportation is terminated (Bublitz et al, 2011; Palmgren & Nissen, 2011). The phosphorylated Asp located in Asp-Lys-Thr-Gly (DKTG) sequence is highly conserved in P-type ATPases. The consensus residue region has been formulized as DKTG[T,S][L,I,V,M][T,I]. The A domain has Asp-Gly-Glu-Thr (DGET) as P4-ATPase–specific signature residues that facilitate dephosphorylation. Besides, two other conserved sequences, Thr-Gly-Asp-Asn (TGDN) and Gly-Asp-Gly-x-Asn-Asp (GDGxND), are located in the P domain that bind $Mg^{2+}$ and connect the ATP-binding region to the transmembrane segments

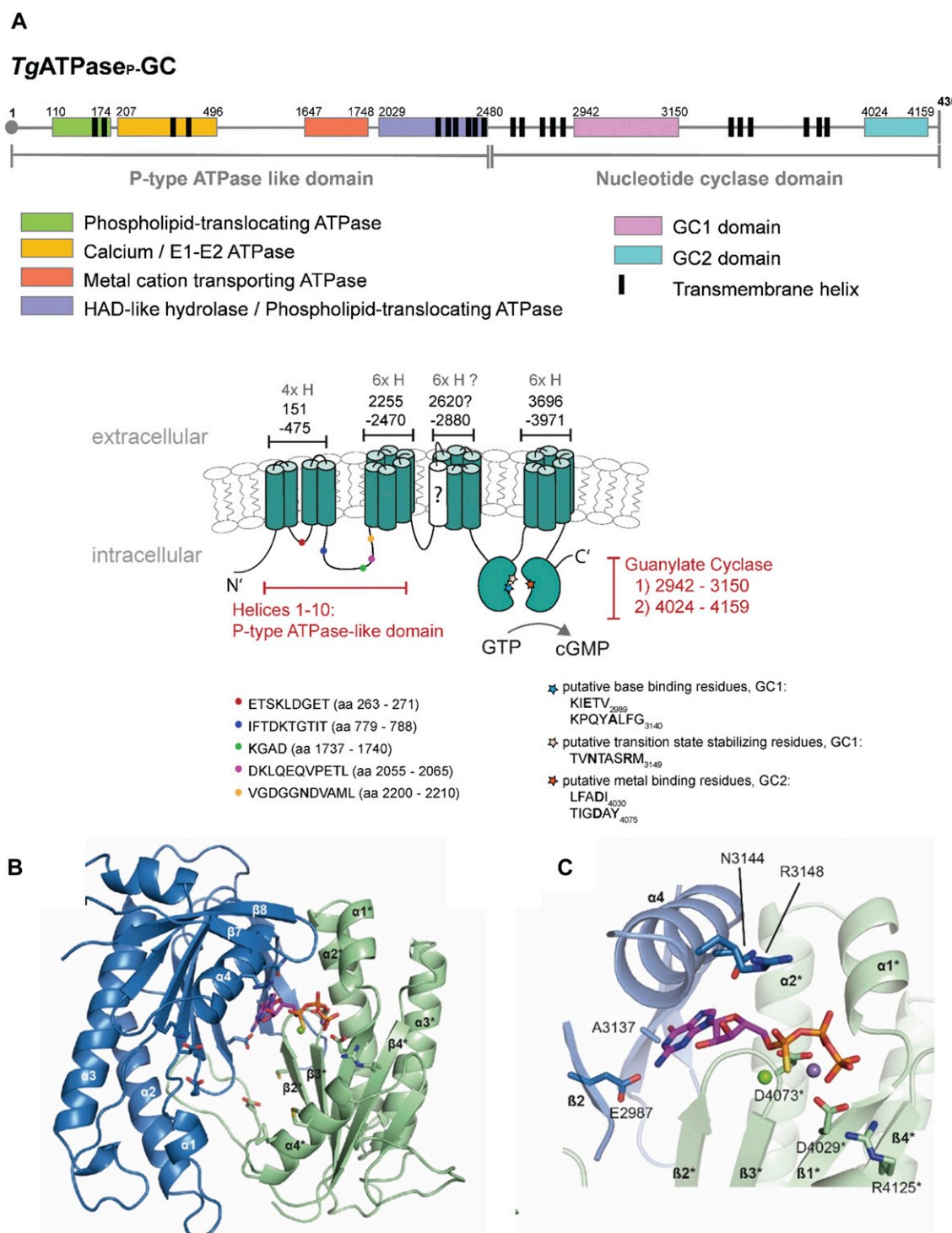

**Figure 1. The genome of *T. gondii* harbors an unusual heterodimeric GC conjugated to P-type ATPase domain.**
**(A)** The primary and secondary topology of *Tg*ATPase$_P$-GC as predicted using TMHMM, SMART, TMpred, Phobius, and NCBI domain search tools. The model was constructed by consensus across algorithms regarding the position of domains and transmembrane spans. The N terminus (1–2,480 aa) containing 10 α-helices resembles P-type ATPase with at least four subdomains (color-coded). The C terminus (2,481–4,367 aa) harbors two potential nucleotide cyclase catalytic regions, termed GC1 and GC2, each following six transmembrane helices. The question-marked (?) helix was predicted only by Phobius (probability score, 752). The color-coded signs on secondary structure show the position of highly conserved sequences in the ATPase and cyclase domains. The key residues involved in the base binding and catalysis of cyclases are also depicted in bold letters. **(B, C)** Tertiary structure of GC1 and GC2 domains based on homology modeling. The ribbon diagrams of GC1 and GC2 suggest a functional activation by pseudo-heterodimerization similar to tmAC. The model shows an antiparallel arrangement of GC1 and GC2, where each domain harbors a seven-stranded β-sheet surrounded by three α-helices. The image in *panel* (C) illustrates a GC1-GC2 heterodimer interface bound to GTPαS. The residues of GC2 labeled with asterisk (*) interact with the phosphate backbone of the nucleotide.

(Axelsen & Palmgren, 1998). T domain includes the ion-binding site, which has a conserved proline in Pro-Glu-Gly-Leu (PEGL) sequence, located usually between TMH4 and TMH5 (Andersen et al, 2016).

The alignment of ATPase domains from $Tg$ATPase$_P$-GC, $Pf$GC$\alpha$, and $Pf$GC$\beta$ with five different members of human P4-ATPases revealed several conserved residues (Fig S2). For example, the second subdomain of $Tg$ATPase$_P$-GC (defined as Ca$^{2+}$-ATPase, yellow colored in Fig 1A) carries DGET signature of the A domain albeit with one altered residue in the region (ETS**K**LDGET instead of ETSNLDGET). $Pf$GC$\alpha$ contains two amino acid mutations in the same region (ETS**LLN**GET) when compared with the human ATP8A1, which translocates phosphatidylserine as its main substrate (Lee et al, 2015). Another replacement (Ser to Thr$^{781}$) was observed both in $Tg$ATPase$_P$-GC and $Pf$GC$\alpha$ at the IF**T**DKTGTIT motif, which harbors the consensus phosphorylated aspartate residue (D) in the P domain. The nucleotide-binding sequence, KGAD in the N domain (third region indicated as cation-ATPase)—the most conserved signature among P-type ATPases—is preserved in $Tg$ATPase$_P$-GC but substituted by a point mutation in $Pf$GC$\alpha$ (Ala to Ser$^{1739}$, KG**S**D). Additional mutation (D to E) was detected in the DKLQ**E**QVPETL sequence located in the last ATPase subdomain of $Tg$ATPase$_P$-GC (highlighted with blue in Fig 1A). Not least, the GDGxND signature is conserved in $Tg$ATPase$_P$-GC but degenerated in $Pf$GC$\alpha$ (Fig S2). Notably, most signature residues could not be identified in $Pf$GC$\beta$, signifying a degenerated ATPase domain. Taken together, our in silico analysis suggests that the N-terminal ATPase domain of $Tg$ATPase$_P$-GC belongs to the P4-ATPase subfamily, and thus likely involved in lipid translocation.

## GC1 and GC2 domains of $Tg$ATPase$_P$-GC form a pseudo-heterodimer GC

The arrangement and architecture of GC1 and GC2 domains in $Tg$ATPase$_P$-GC correspond to mammalian transmembrane adenylate cyclase (tmAC) of the class III (Linder & Schultz, 2003). The latter is activated by G-proteins to produce cAMP after extracellular stimuli (e.g., hormones). The cyclase domains of tmACs, C1, and C2 form an antiparallel pseudo-heterodimer with one active and one degenerated site at the dimer interface (Linder & Schultz, 2003). Amino acids from both domains contribute to the binding site, and seven conserved residues are identified to play essential roles for nucleotide binding and catalysis (Linder & Schultz, 2003; Sinha & Sprang, 2006; Steegborn, 2014). These include two aspartate residues, which bind two divalent metal cofactors (Mg$^{+2}$, Mn$^{+2}$) crucial for substrate placement and turnover. An arginine and asparagine stabilize the transition state, although yet another arginine binds the terminal phosphate (P$\gamma$) of the nucleotide. A lysine/aspartate pair underlies the selection of ATP over GTP as the substrate. By contrast, a glutamate/cysteine or glutamate/alanine pair defines the substrate specificity as GTP in GCs. Nucleotide binding and transition-state stabilization are conferred by one domain, whereas the other domain by direct or via the interaction of bound metal ions with the phosphates of the nucleotide in tmACs (Linder, 2005; Sinha & Sprang, 2006; Steegborn, 2014).

The sequence alignment of GC1 and GC2 domains from $Tg$ATPase$_P$-GC to their orthologous GCs/ACs showed that GC1 contains a 74-residue-long loop insertion (3,033–3,107 aa), unlike other cyclases (Fig S3). A shorter insertion (~40 aa) was also found in $Pf$GC$\alpha$. The tertiary model structure (depleted for the loop inserted between $\alpha$3 and $\beta$4 of GC1) shows that both domains consist of a seven-stranded $\beta$-sheet surrounded by three helices (Fig 1B and C). The key functional amino acid residues with some notable substitutions could be identified as distributed across GC1 and GC2. In GC1 domain, one of the two metal-binding (Me) aspartates is replaced by glutamate (E2991), whereas both are conserved in GC2 (D4029 and D4073) (Figs 1C and S3). The transition-state stabilizing (Tr) asparagine (N3144) and arginine (R3148) residues are located within the GC1 domain (Fig 1C); however, both are replaced by leucine (L4153) and methionine (M4157), respectively, in the GC2 domain (Fig S3). Another arginine (R4125) that is responsible for phosphate binding (P$\gamma$) in tmACs is conserved in the GC2 domain (Fig 1C), whereas it is substituted by K3116 in GC1 (Fig S3).

The cyclase specificity defining residues (B) are glutamate/alanine (E2987/A3137) and cysteine/aspartate (C4069/D4146) pairs in GC1 and GC2, respectively (Figs 1C and S3). The E/A identity of the nucleotide-binding pair in GC1 is indicative of specificity towards GTP. Thus, we propose that GC1 and GC2 form a pseudo-heterodimer and function as a guanylate cyclase (Fig 1C). Similar to tmACs, one catalytically active and one degenerated site are allocated at the dimer interface to make $Tg$ATPase$_P$-GC functional. However, the sequence of GC1 and GC2 is inverted in $Tg$ATPase$_P$-GC, which means that, unlike tmAC, GC1 domain contributes to the nucleotide and transition-state binding residues of the active site, whereas GC2 harbors two aspartates crucial for metal ion binding. Although our multiple attempts to test the recombinant activity of GC1, GC2, and GC1 fused to GC2 (GC1+GC2) were futile (Fig S4), we were able to show the involvement of $Tg$ATPase$_P$-GC in cGMP synthesis by mutagenesis studies in tachyzoites (discussed below, Fig 4).

## Overexpression and purification of recombinant GC1 and GC2 domains

With an objective to determine the functionality of $Tg$ATPase$_P$-GC, we expressed ORFs of GC1 (M$^{2850}$-S$^{3244}$), GC2 (M$^{3934}$-Q$^{4242}$), and GC1+GC2 (M$^{2850}$-Q$^{4242}$) in *Escherichia coli* (Fig S4A). Positive clones were verified by PCR screening and sequencing (Fig S4B). An overexpression of GC1 and GC2 as 6x His-tagged in the M15 strain resulted in inclusion bodies, which did not allow us to use native conditions to purify proteins. We nevertheless purified them through an Ni-NTA column under denaturing conditions. Purified GC1 and GC2 exhibited an expected molecular weight of 47 and 38 kD, respectively (Fig S4C). Our attempts to purify GC1+GC2 protein were futile, however. To test the catalytic activity of purified GC1 and GC2 domains, we executed an in vitro GC assay. Neither for GC1 nor for GC2 was any functionality detected when tested separately or together. Further optimization of the protein purification process yielded no detectable GC activity. The GC assay was also performed with the bacterial lysates expressing specified domains; however, no cGMP production was observed, as judged by high-performance liquid chromatography (Fig S4D).

Furthermore, we examined whether GC1 and GC2 can function as adenylate cyclase using a bacterial complementation assay, as

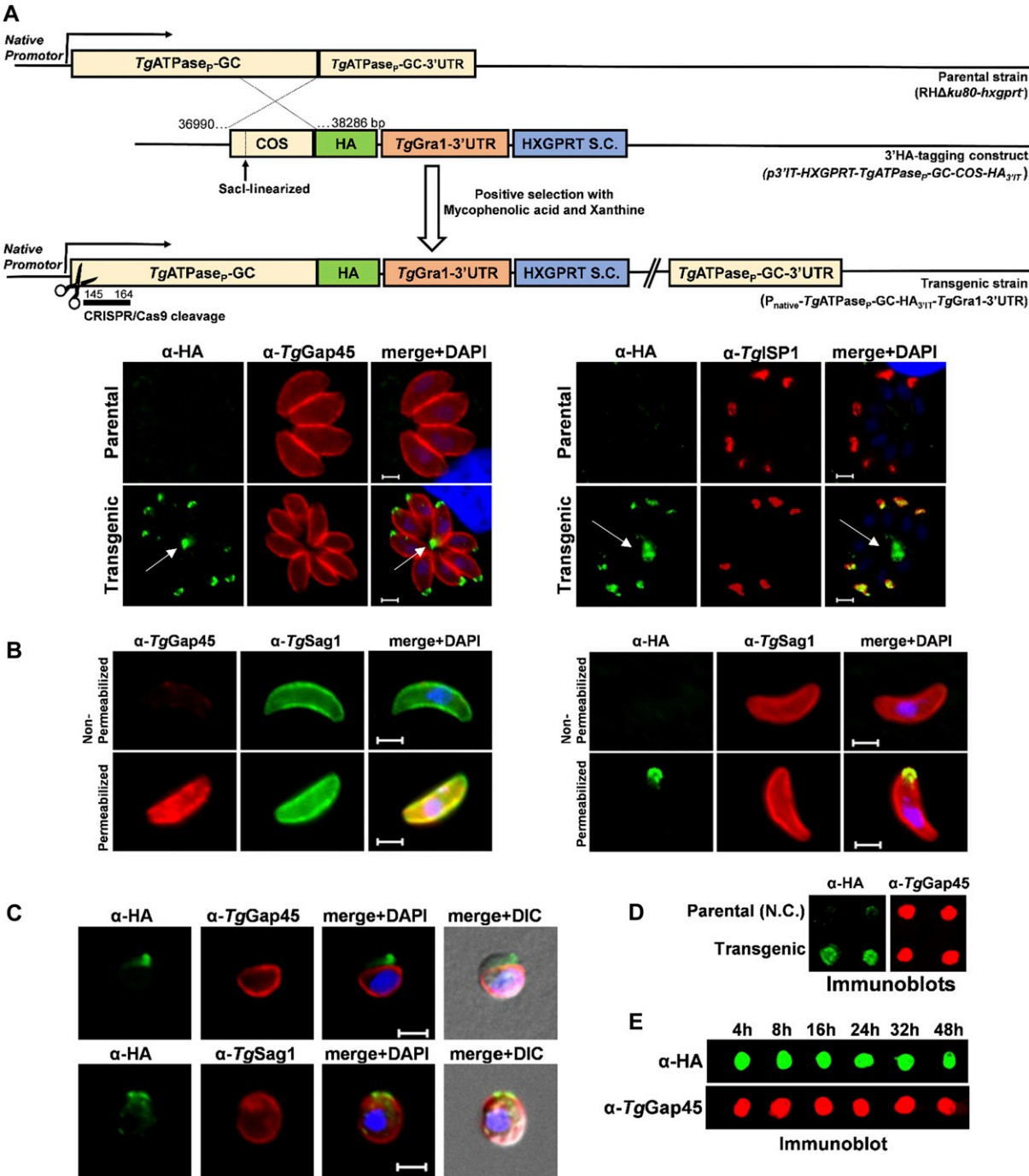

**Figure 2. *Tg*ATPase_P-GC is a constitutively expressed protein located at the apical end in the plasma membrane of *T. gondii*.**
**(A)** Scheme for the genomic tagging of *Tg*ATPase_P-GC with a 3'-end HA epitope. The *Sac*I-linearized plasmid for 3'-insertional tagging (*p3'IT-HXGPRT-TgATPase_P-GC-COS-HA_{3'IT}*) was transfected into parental (RHΔ*ku80-hxgprt*⁻) strain followed by drug selection. Intracellular parasites of the resulting transgenic strain (P_native-*Tg*ATPase_P-GC-HA_{3'IT}-*Tg*Gra1-3'UTR) were subjected to staining by specific antibodies (24 h post-infection). *Arrows* indicate the location of the residual body. The host-cell and parasite nuclei were stained by DAPI. Scale bars represent 2 μm. **(B)** Immunofluorescence staining of extracellular parasites expressing *Tg*ATPase_P-GC-HA_{3'IT}. The α-HA immunostaining of the free parasites was performed before or after membrane permeabilization either using PBS without additives or detergent-supplemented PBS with BSA, respectively. The appearance of *Tg*Gap45 signal (located in the IMC) only after permeabilization confirms functionality of the assay. *Tg*Sag1 is located in the plasma membrane and, thus, visible under both conditions. Scale bars represent 2 μm. **(C)** Immunostaining of extracellular parasites encoding *Tg*ATPase_P-GC-HA_{3'IT} after drug-induced splitting of the IMC from the plasma membrane. Tachyzoites were incubated with α-toxin (20 nM, 2 h) before immunostaining with α-HA antibody in combination with primary antibodies recognizing IMC (α-*Tg*Gap45) or plasmalemma (α-*Tg*Sag1), respectively. Scale bars represent 2 μm. **(D, E)** Immunoblots of tachyzoites expressing *Tg*ATPase_P-GC-HA_{3'IT} and of the parental strain (RHΔ*ku80-hxgprt*⁻, negative control). The protein samples prepared from extracellular parasites ($10^7$) were directly loaded onto membrane blot, followed by staining with α-HA and α-*Tg*Gap45 antibodies. Samples in *panel* (E) were collected at different time periods during the lytic cycle and stained with α-HA and α-*Tg*Gap45 (loading control) antibodies. COS, crossover sequence; S.C., selection cassette.

described elsewhere (Karimova et al, 1998) (Fig S4E). GC1, GC2, and GC1+GC2 proteins were expressed in the BTH101 strain of *E. coli*, which is deficient in the adenylate cyclase activity and so unable to use maltose as a carbon source. The strain produced white colonies on MacConkey agar containing maltose, which would otherwise be red-colored upon induction of cAMP-dependent disaccharide catabolism. We observed that unlike the positive control (adenylate cyclase from *E. coli*), BTH101 strains expressing GC1, GC2, or GC1+GC2 produced only white colonies in each case (Fig S4E), which could either be attributed to inefficient expression or a lack of adenylate cyclase activity in accord with the presence of signature residues defining the specificity for GTP in indicated domains (Fig S3).

Notwithstanding technical issues with our expression model or enzyme assay, it is plausible that the P4-ATPase domain is required for the functionality of cyclase domains in $Tg$ATPase$_P$-GC, as also suggested by two recent studies (Brown & Sibley, 2018; Bisio et al, 2019). In similar experiments conducted with *Plasmodium* GCs, $Pf$GC$\alpha$, and $Pf$GC$\beta$, the GC activity could only be confirmed for $Pf$GC$\beta$ but not for $Pf$GC$\alpha$ (Carucci et al, 2000), which happens to be the nearest ortholog of $Tg$ATPase$_P$-GC (refer to phylogeny in Fig S1).

### $Tg$ATPase$_P$-GC is constitutively expressed in the plasma membrane at the apical pole

To gain insight into the endogenous expression and localization of $Tg$ATPase$_P$-GC protein, we performed epitope tagging of the gene in tachyzoites of *T. gondii* (Fig 2A). The parental strain was transfected with a plasmid construct allowing 3'-insertional tagging of $Tg$ATPase$_P$-GC with a HA tag by single homologous crossover. The resulting transgenic strain (P$_{native}$-$Tg$ATPase$_P$-GC-HA$_{3'IT}$-$Tg$Gra1-3'UTR) encoded HA-tagged $Tg$ATPase$_P$-GC under the control of its native promoter. Notably, the fusion protein localized predominantly at the apex of the intracellularly growing parasites, as construed by its costaining with $Tg$Gap45, a marker of the inner membrane complex (IMC) (Gaskins et al, 2004) (Fig 2A, *left*). The apical location of $Tg$ATPase$_P$-GC-HA$_{3'IT}$ was confirmed by its colocalization with the IMC sub-compartment protein 1 ($Tg$ISP1) (Beck et al, 2010) (Fig 2A, *right*). Moreover, we noted a significant expression of $Tg$ATPase$_P$-GC-HA$_{3'IT}$ outside the parasite periphery within the residual body (Fig 2A, marked with *arrows*), which has also been observed for several other proteins, such as Rhoptry Neck 4 (RON4) (Bradley et al, 2005). To assess the membrane location and predicted C-terminal topology of the protein, we stained extracellular parasites with $\alpha$-HA antibody before and after detergent permeabilization of the parasite membranes (Fig 2B). The HA staining and apical localization of $Tg$ATPase$_P$-GC were detected only after the permeabilization, indicating that the C terminus of $Tg$ATPase$_P$-GC faces the parasite interior, as shown in the model (Fig 1A).

We then treated the extracellular parasites with $\alpha$-toxin to separate the plasma membrane from IMC and thereby distinguish the distribution of $Tg$ATPase$_P$-GC-HA$_{3'IT}$ between both entities. By staining of tachyzoites with two markers, that is, $Tg$Gap45 for the IMC and $Tg$Sag1 for the PM, we could show an association of $Tg$ATPase$_P$-GC-HA$_{3'IT}$ with the plasma membrane (Fig 2C). Making of a transgenic line encoding $Tg$ATPase$_P$-GC-HA$_{3'IT}$ also enabled us to evaluate its expression pattern by immunoblot analysis

throughout the lytic cycle, which recapitulates the successive events of gliding motility, host-cell invasion, intracellular replication, and egress leading to host-cell lysis. $Tg$ATPase$_P$-GC is a bulky protein (477-kD) with several transmembrane regions; hence, it was not possible for us to successfully resolve it by gel electrophoresis and transfer onto nitrocellulose membrane for immunostaining. We nonetheless performed the dot blot analysis by loading protein samples directly onto an immunoblot membrane (Fig 2D and E). Unlike the parental strain (negative control), which showed only a faint (background) $\alpha$-HA staining, we observed a strong signal in the $Tg$ATPase$_P$-GC-HA$_{3'IT}$–expressing strain (Fig 2D). Samples of transgenic strain collected at various periods embracing the entire lytic cycle indicated a constitutive and steady expression of $Tg$ATPase$_P$-GC-HA$_{3'IT}$ in tachyzoites (Fig 2E).

### $Tg$ATPase$_P$-GC is essential for the parasite survival

Having established the expression profile and location, we next examined the physiological importance of $Tg$ATPase$_P$-GC for tachyzoites. Our multiple efforts to knockout the $Tg$ATPase$_P$-GC gene by double homologous recombination were unrewarding, suggesting its essentiality during the lytic cycle (lethal phenotype). We, therefore, used the strain expressing $Tg$ATPase$_P$-GC-HA$_{3'IT}$ to monitor the effect of genetic disruption immediately after the plasmid transfection (Fig 3). To achieve this, we executed a CRISPR/Cas9–directed cleavage in $Tg$ATPase$_P$-GC gene and then immunostained parasites at various periods to determine a time-elapsed loss of HA signal (Fig 3A). Within a day of transfection, about 4% of vacuoles had lost the apical staining of $Tg$ATPase$_P$-GC-HA$_{3'IT}$ (Fig 3B). The number of vacuoles without the HA signal remained constant until the first passage (P1, 24–40 h). However, the parasite growth reduced gradually during the second passage (P2, 72–88 h) and fully seized by the third passage (P3, 120–136 h) (Fig 3B). The same assay also allowed us to quantify the replication rates of HA-negative parasites in relation to the HA-positive parasites by counting their numbers in intracellular vacuoles (Fig 3C). As expected, the fraction of small vacuoles comprising just one or two parasites was much higher in the nonexpression (HA-negative) parasites. Inversely, the progenitor strain expressing $Tg$ATPase$_P$-GC-HA$_{3'IT}$ showed predominantly a higher percentage of bigger vacuoles with 16–64 parasites. By the third passage, we detected only single-parasite vacuoles in the mutant, demonstrating an essential role of $Tg$ATPase$_P$-GC for the asexual reproduction.

### Genetic repression of $Tg$ATPase$_P$-GC blights the lytic cycle

Although an indispensable nature of $Tg$ATPase$_P$-GC for tachyzoites could be established, the above strategy did not yield us a clonal mutant for in-depth biochemical and phenotypic analyses because of an eventually mortal phenotype. Hence, we engineered another parasite strain expressing $Tg$ATPase$_P$-GC-HA$_{3'IT}$, in which the native 3'UTR of the gene was flanked with two loxP sites (Fig 4A). Cre recombinase–mediated excision of the 3'UTR combined with a negative selection, as reported earlier (Brecht et al, 1999), permitted down-regulation of $Tg$ATPase$_P$-GC-HA$_{3'IT}$. Genomic screening using specific primers confirmed a successful generation of the mutant

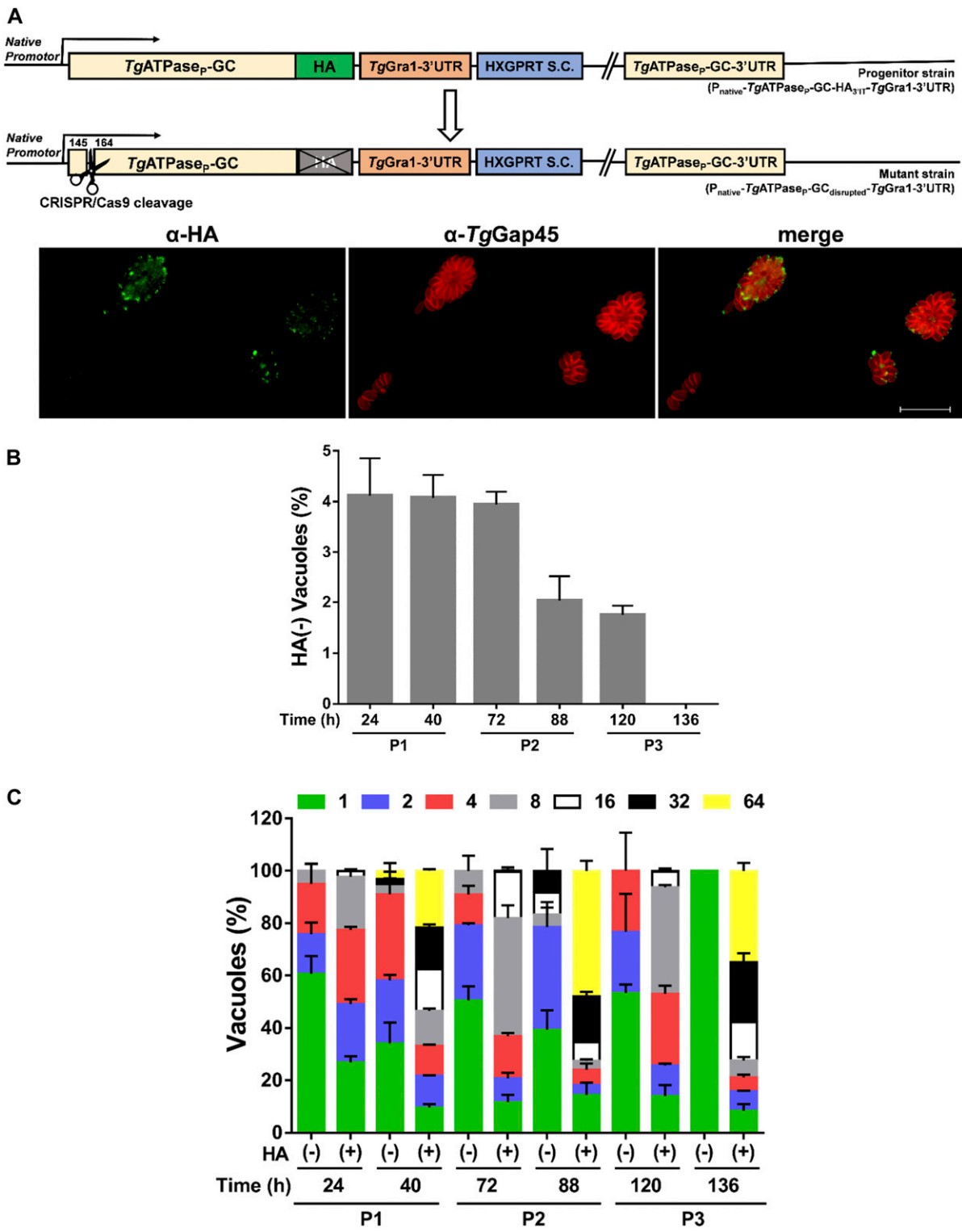

**Figure 3. Genetic disruption of TgATPase$_P$-GC is lethal to tachyzoites of *T. gondii*.**
**(A)** Scheme for the CRISPR/Cas9-mediated disruption of the gene in parasites expressing TgATPase$_P$-GC-HA$_{3'IT}$. The guide RNA was designed to target the nucleotides between 145 and 164 bp in the progenitor strain (P$_{native}$-TgATPase$_P$-GC-HA$_{3'IT}$-TgGra1-3'UTR). Image shows the loss of HA signal in some vacuoles after CRISPR/Cas9-cleavage. Parasites were transfected with the *pU6-TgATPase$_P$-GC$_{sgRNA}$-Cas9* vector and then stained with α-HA and α-TgGap45 antibodies at specified periods. Scale bars represent 20 μm. **(B)** Quantitative illustration of TgATPase$_P$-GC-HA$_{3'IT}$–disrupted mutant parasites from *panel* (A). The HA-negative vacuoles harboring at least two parasites were scored during successive passages (P1–P3). **(C)** The replication rates of the HA$^+$ and HA$^-$ tachyzoites, as evaluated by immunostaining (*panel A*). About 500–600 vacuoles were enumerated for the parasite numbers per vacuole (n = 3 assays).

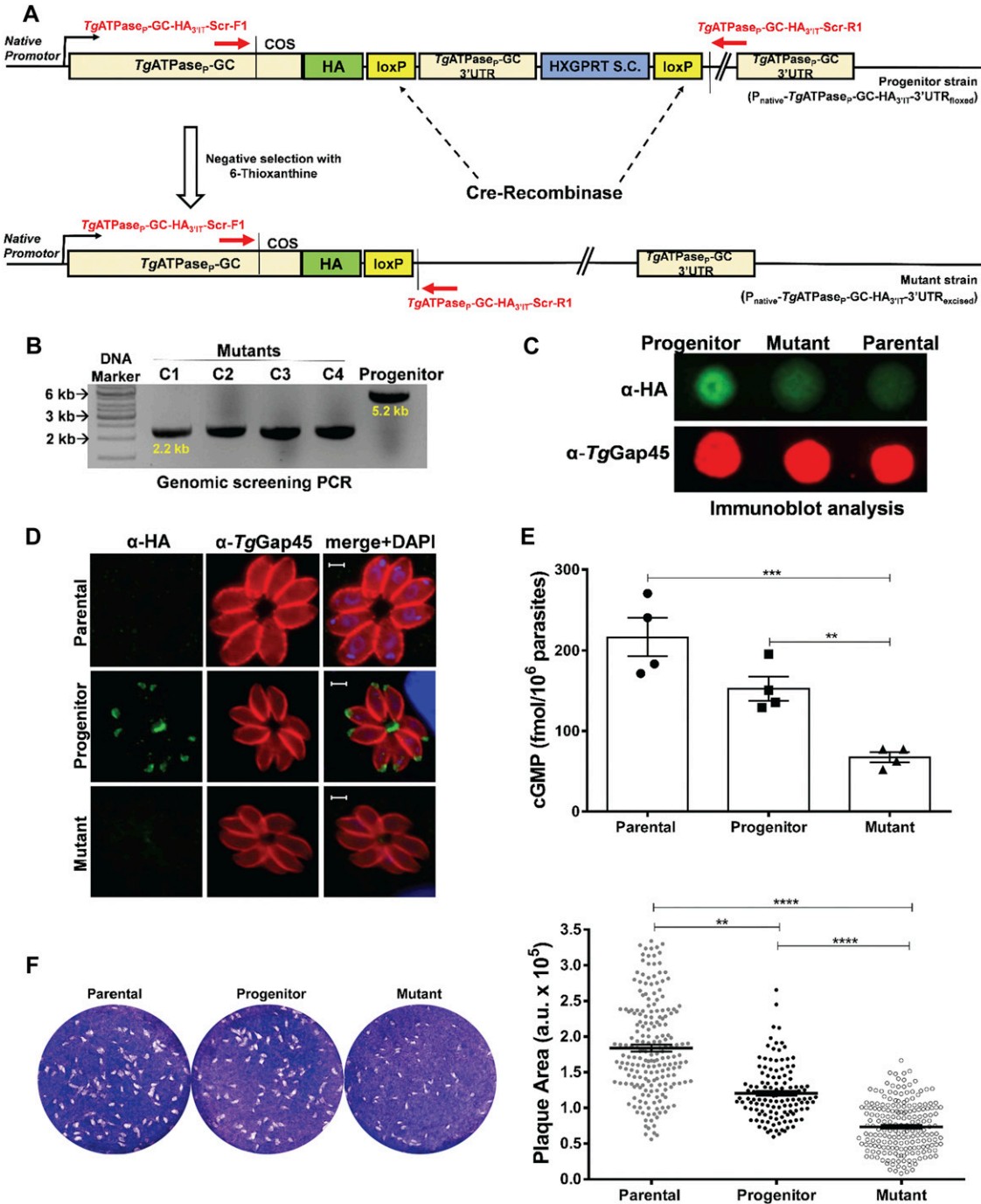

**Figure 4. Cre recombinase–mediated down-regulation of cGMP synthesis impairs the lytic cycle of _T. gondii_.**
**(A)** Schematics for making the parasite mutant (P$_{native}$-_Tg_ATPase$_P$-GC-HA$_{3'IT}$-3'UTR$_{excised}$). A vector expressing Cre recombinase was transfected into the progenitor strain (P$_{native}$-_Tg_ATPase$_P$-GC-HA$_{3'IT}$-3'UTR$_{floxed}$), in which 3'UTR of _Tg_ATPase$_P$-GC was flanked with Cre/loxP sites. Parasites transfected with a vector expressing Cre recombinase were selected for the loss of HXGPRT selection cassette (S.C.) using 6-thioxanthine. **(B)** Genomic screening of the _Tg_ATPase$_P$-GC mutant confirming Cre-mediated excision of 3'UTR and HXGPRT. Primers, indicated as red _arrows_ in _panel_ (A), were used to PCR-screen the gDNA isolated from four different mutant clones (C1–C4) along with the progenitor strain. **(C)** Immunoblot showing repression of _Tg_ATPase$_P$-GC-HA$_{3'IT}$ in parasites with excised 3'UTR with respect to the progenitor and parental (RHΔ_ku80-hxgprt_⁻) strains. Parasites (10⁷) were subjected to the dot blot analysis using α-HA and α-_Tg_Gap45 (loading control) antibodies. **(D)** Immunostaining of the mutant (_Tg_ATPase$_P$-GC-HA$_{3'IT}$-3'UTR$_{excised}$) and progenitor parasites revealing loss of HA signal in the former strain. Parental strain was used as a negative control for the background staining. Parasites were stained with α-HA and α-_Tg_Gap45 antibodies 24 h postinfection. Scale bars represent 2 μm. **(E)** Changes in the steady-state cGMP level of the mutant compared with the parental and progenitor strains. Fresh syringe-released parasites (5 × 10⁶) were subjected to ELISA-based cGMP measurements (n = 4 assays). **(F)** Plaque assays using the _Tg_ATPase$_P$-GC mutant, progenitor, and parental strains. The dotted white areas and blue staining signify plaques and intact host-cell monolayers, respectively (_left_). The area of each plaque (arbitrary units, a. u.) embodies the growth fitness of indicated strains. 150–200 plaques of each strain were evaluated (_right_) from three assays. **P ≤ 0.01; ****P ≤ 0.0001.

($P_{native}$-$Tg$ATPase$_P$-GC-HA$_{3'IT}$-3'UTR$_{excised}$), which yielded a 2.2-kb amplicon as opposed to 5.2-kb in the progenitor strain ($P_{native}$-$Tg$ATPase$_P$-GC-HA$_{3'IT}$-3'UTR$_{floxed}$) (Fig 4B). Immunoblots of a clonal mutant showed an evident repression of the protein (Fig 4C). Densitometric analysis of $Tg$ATPase$_P$-GC-HA$_{3'IT}$ revealed about 65% reduction in the mutant compared with the progenitor strain. Knockdown was further endorsed by loss of HA-staining in immunofluorescence assay (IFA) (Fig 4D), where about 94% vacuoles lost their signal and the rest (~6%) displayed only a faint or no HA signal (Fig S5).

Next, we evaluated if repression of $Tg$ATPase$_P$-GC translated into declined cGMP synthesis by the parasite. Indeed, we measured ~60% regression in the steady-state levels of cGMP in the mutant (Fig 4E), equating to the decay at the protein level (Fig 4C and D). We then measured the comparative fitness of the mutant, progenitor, and parental strains by plaque assays (Fig 4F). As anticipated, the mutant exhibited about 65% and 35% reduction in plaque area when compared with the parental and progenitor strains, respectively, which correlated rather well with the residual expression of $Tg$ATPase$_P$-GC in the immunoblot as well as cGMP assays (Fig 4C–E). The progenitor strain also showed ~30% impairment corresponding to reduction in its cGMP level compared with the parental strain that is likely due to epitope tagging and introduction of loxP sites between the last gene exon and 3'UTR (Fig 4A). These data together with the above results show that $Tg$ATPase$_P$-GC functions as a GC, and its catalytic activity is necessary for the lytic cycle.

### $Tg$ATPase$_P$-GC regulates multiple events during the lytic cycle

The availability of an effective mutant encouraged us to study the importance of $Tg$ATPase$_P$-GC for discrete steps of the lytic cycle, including invasion, cell division, egress, and gliding motility (Fig 5). The replication assay revealed a modestly higher fraction of smaller vacuoles with two parasites in early culture (24 h) of the mutant compared with the control strains; the effect was assuaged at a later stage (40 h), however (Fig 5A, left). The average parasite numbers in each vacuole was also scored to ascertain these data. Indeed, no significant difference was seen in numbers of the $Tg$ATPase$_P$-GC mutant with respect to the control strains in the late-stage culture, even though a slight delay was observed in the early culture (Fig 5A, right). Two recent studies depleting $Tg$ATPase$_P$-GC using different methods (Brown & Sibley, 2018; Yang et al, 2019) also concluded that the protein is not required for parasite replication. We quantified about 30% decline in the invasion efficiency of the mutant down from 80 to 53% (Fig 5B). Hence, a minor replication defect at 24 h may be a consequence of poor host-cell invasion by the parasite. The effect of protein repression was more pronounced in egress assay, where the mutant showed 70% decline in natural egress when compared with the parental strain and 40% defect in relation to the progenitor strain (40–48 h postinfection, Fig 5C), as also shown by others (Brown & Sibley, 2018; Bisio et al, 2019; Yang et al, 2019). Notably though, the egress defect was not apparent upon prolonged (64 h) culture. Such compensation at a later stage is probably caused by alternative (calcium dependent protein kinase) signaling cascades (Lourido et al, 2012)—a notion also reflected in the study of Yang et al (2019), where $Tg$ATPase$_P$-GC deficiency could be compensated by Ca$^{2+}$ ionophore.

Because invasion and egress are mediated by gliding motility (Frénal et al, 2017), we tested our mutant for the latter phenotype. We determined that the average motile fraction was reduced by more than half in the mutant, and trail lengths of moving parasites were remarkably shorter (~18 $\mu m$) than the control strain (~50 $\mu m$) (Fig 5D). Not least, as witnessed in plaque assays (Fig 4F), we found a steady, albeit not significant, decline in the invasion and egress rates of the progenitor when compared with the parental strain, which further confirms a correlation across all phenotypic assays. A partial phenotype prompted us to pharmacologically inhibit the residual cGMP signaling via PKG in the $Tg$ATPase$_P$-GC mutant. We used compound 2 (C2), which has been shown to block mainly $Tg$PKG but also calcium-dependent protein kinase 1 (Donald et al, 2006). As rationalized, C2 treatment subdued the gliding motility of the mutant as well as of the progenitor strain (Fig 5E). The impact of C2 was accentuated in both strains, likely because of cumulative effect of genetic repression and drug inhibition. Inhibition was stronger in the $Tg$ATPase$_P$-GC mutant than the progenitor strain that can be attributed to a potentiated inhibition of the residual cGMP signaling in the knockdown strain.

### Phosphodiesterase inhibitors can rescue defective phenotypes of $Tg$ATPase$_P$-GC mutant

To further validate our findings, we deployed two inhibitors of cGMP-specific PDEs, namely, zaprinast and BIPPO, which are known to inhibit parasite enzymes along with human PDE5 and PDE9, respectively (Yuasa et al, 2005; Howard et al, 2015). We reasoned that drug-mediated elevation of cGMP could mitigate the phenotypic defects caused by deficiency of GC in the mutant. As shown (Fig S6A), both drugs led to a dramatic increase in the motile fraction and trail lengths of the progenitor and $Tg$ATPase$_P$-GC–mutant strains. The latter parasites were as competent as the former after the drug exposure. A similar restoration of phenotype in the mutant was detected in egress assays; the effect of BIPPO was much more pronounced than zaprinast, leading to egress of nearly all parasites (Fig S6B). In contrast to the motility and egress, a treatment of BIPPO and zaprinast resulted in a surprisingly divergent effect on the invasion rates of the two strains (Fig S6C). BIPPO exerted an opposite effect, i.e., a reduction in invasion of the progenitor and mutant. Impairment was stronger in the former strain; hence, we noted a reversal of the phenotype when compared with the control samples. A fairly similar effect was seen with zaprinast, although it was much less potent than BIPPO, as implied previously (Howard et al, 2015). These observations can be attributed to differential elevation of cGMP and possibly cAMP (above certain threshold) caused by PDE inhibitors, which inhibits the host-cell invasion but promotes the parasite motility and egress.

### Genetic knockdown of $Tg$PKG phenocopies the attenuation of $Tg$ATPase$_P$-GC

To consolidate the aforesaid work on $Tg$ATPase$_P$-GC, we implemented the same genomic tagging, knockdown, and phenotyping approaches to $Tg$PKG (Figs S7 and 6). Briefly, a parasite strain expressing $Tg$PKG with a C-terminal HA-tag under the control of endogenous regulatory

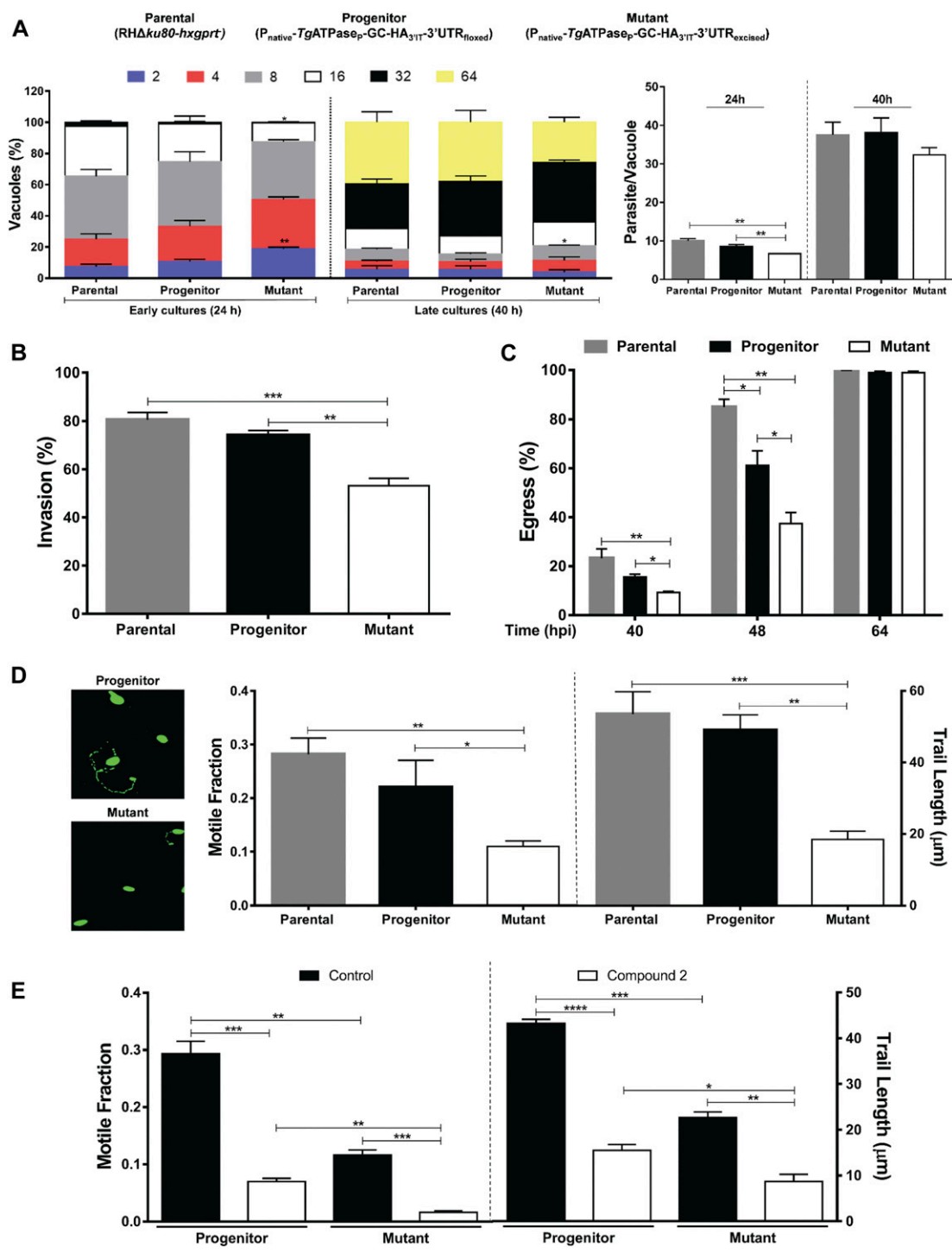

**Figure 5. cGMP signaling governs the key events during the lytic cycle of *T. gondii*.**
**(A–D)** In vitro phenotyping of the *Tg*ATPase$_P$-GC mutant, its progenitor, and parental strains. The intracellular replication (A), host-cell invasion (B), parasite egress (C), and gliding motility (D) were assessed using standard phenotyping methods. The progenitor and mutant strains were generated as shown in Figs 2A and 4A, respectively. The replication rates were analyzed 24 and 40 h postinfection by scoring the parasite numbers in a total of 500–600 vacuoles after staining with *α-Tg*Gap45 antibody (*panel A, left*) (n = 4 assays). The average parasite numbers per vacuole is also depicted (*panel A, right*). Invasion and egress rates were calculated by dual staining with *α-Tg*Gap45 and *α-Tg*Sag1 antibodies. In total, 1,000 parasites of each strain from four assays were examined to estimate the invasion efficiency. The natural egress of tachyzoites was measured after 40, 48, and 64 h by scoring 500–600 vacuoles of each strain (n = 3 assays). To estimate the gliding motility, fluorescent images stained with *α-Tg*Sag1 antibody were analyzed for the motile fraction (500 parasites of each strain), and 100–120 trail lengths per strain were measured (n = 3 assays). **(E)** Effect of PKG inhibitor compound 2 (2 *μ*M) on the motility of *Tg*ATPase$_P$-GC mutant and its progenitor strain (500 parasites of each strain, n = 3 assays). A total of 100 trails in the progenitor, and 15 trails of the mutant (due to severe defect) were measured. *$P \leq 0.05$; **$P \leq 0.01$; ***$P \leq 0.001$; and ****$P \leq 0.0001$.

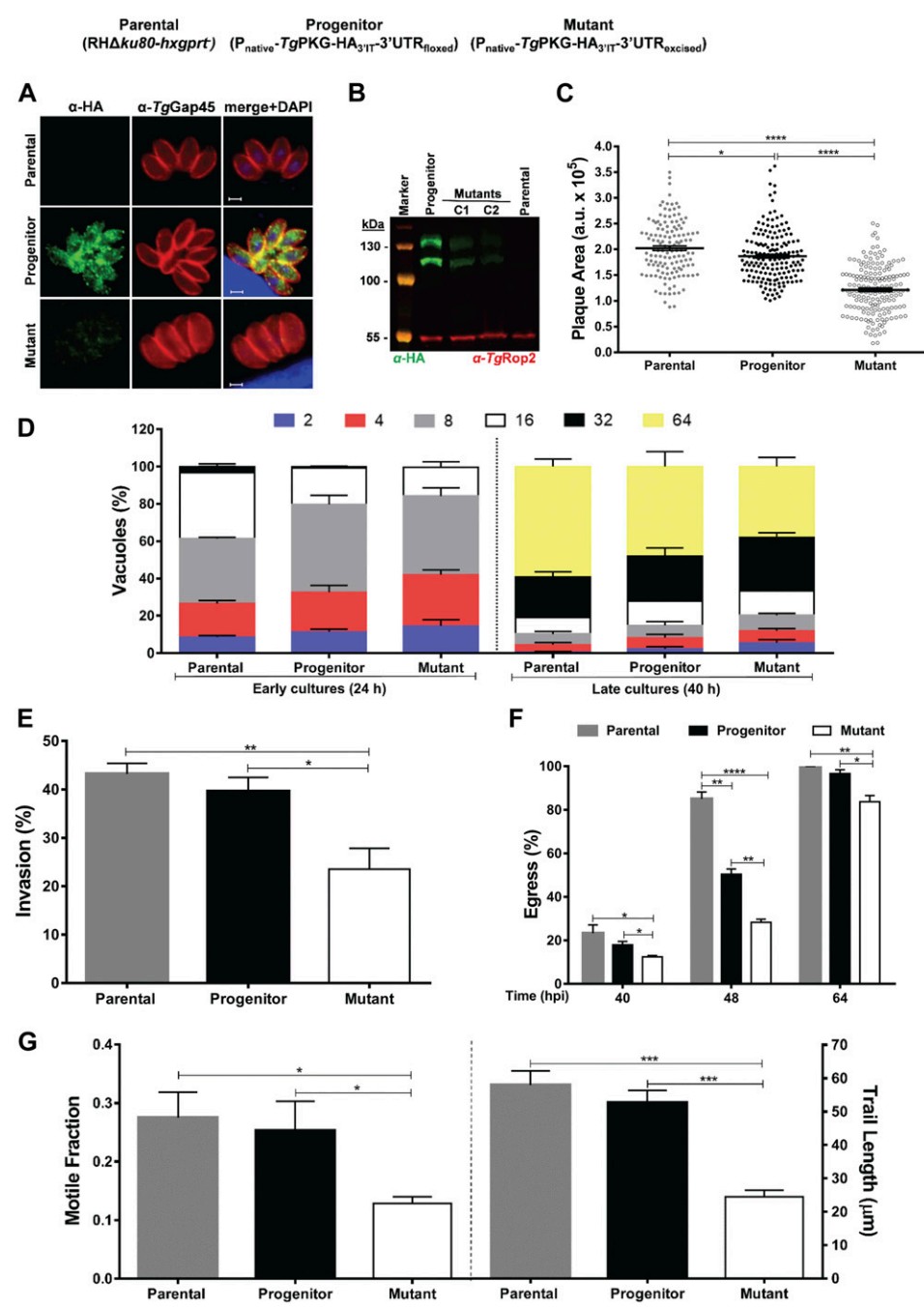

**Figure 6. Mutagenesis of *Tg*PKG recapitulates the phenotype of the *Tg*ATPase_P-GC mutant.**

**(A–F)** Phenotyping of the *Tg*PKG mutant in comparison with its progenitor and parental strains. For making of the mutant, refer to Fig S7. **(A)** Images demonstrating the expression of *Tg*PKG-HA_{3'IT} in the progenitor strain, and its down-regulation in the mutant. The parental strain served as a negative control. Intracellular parasites (24 h postinfection) were stained with α-HA and α-*Tg*Gap45 antibodies. The merged image includes DAPI-stained host and parasite nuclei in blue. Scale bars represent 2 μm. **(B)** Immunoblot depicting the expression of *Tg*PKG isoforms in clonal mutants (C1 and C2) along with the progenitor and parental strains. Extracellular tachyzoites (2 × 10^7) of each strain were subjected to protein isolation followed by immunoblotting with α-HA antibody. Expression of 112 and 135-kD isoforms in the progenitor and mutants, but not in the parental strain, confirms efficient 3'-HA tagging and successful knockdown of the protein, respectively. *Tg*Rop2 served as loading control. **(C)** Plaque assay revealing comparative growth of the mutant, progenitor, and parental strains. The plaque area is shown in arbitrary units (a. u.). A total of 140–170 plaques of each strain were scored from three assays. **(D)** Replication rates of indicated strains during early (24 h) and late (40 h) cultures. Tachyzoites proliferating in their vacuoles were stained with α-*Tg*Gap45 antibody and counted from 400 to 500 vacuoles for each strain (n = 3 assays). **(E, F)** Invasion and egress of the designated parasites as judged by dual-color staining. Intracellular tachyzoites were immunostained red using α-*Tg*Gap45 antibody, whereas extracellular ones appeared two-colored (red and green), stained with both α-*Tg*Gap45 and α-*Tg*Sag1 antibodies. In total, 1,000–1,200 parasites were evaluated to score the invasion rate of each strain (n = 5 assays). The percentage of ruptured vacuoles at indicated periods was determined by observing 400–500 vacuoles of each strain from three experiments. **(G)** Motile fraction and trail lengths of the indicated parasite strains. Extracellular parasites immunostained with α-*Tg*Sag1 were analyzed for the motile fraction (600 parasites) and trail lengths (100 parasites) (n = 3 assays). *P ≤ 0.05; **P ≤ 0.01; ***P ≤ 0.001; and ****P ≤ 0.0001.

elements, and floxed 3′UTR and hypoxanthine-xanthine-guanine phosphoribosyltransferase (HXGPRT) selection cassette was generated by 3′-insertional tagging strategy (Fig S7A). The resultant progenitor strain (P$_{native}$-*Tg*PKG-HA$_{3′IT}$-3′UTR$_{floxed}$) revealed *Tg*PKG-HA$_{3′IT}$ expression in the cytoplasm and membranes of intracellular parasites (Fig 6A), as reported earlier (Donald et al, 2002; Gurnett et al, 2002; Brown et al, 2017). Moreover, immunoblot demonstrated two isoforms of *Tg*PKG in the progenitor strain, where the expression of smaller isoform (112-kD) was stronger than the larger one (135-kD). We then executed Cre-mediated knockdown of *Tg*PKG-HA$_{3′IT}$ by excising the loxP-flanked 3′UTR in the progenitor strain (Fig S7A). As intended, genomic screening with specified primers yielded 1.9-kb amplicons in the isolated clonal mutants (P$_{native}$-*Tg*PKG-HA$_{3′IT}$-3′UTR$_{excised}$) as opposed to a 4.8-kb band in the progenitor strain (P$_{native}$-*Tg*PKG-HA$_{3′IT}$-3′UTR$_{floxed}$), which confirmed the excision of 3′UTR and HXGPRT marker (Fig S7B). The efficiency of Cre-loxP recombination was about 93%, as judged by scoring HA$^+$ and HA$^-$ vacuoles (Fig S7C). A repression of *Tg*PKG-HA$_{3′IT}$ was validated by immunofluorescence and immunoblot assays (Fig 6A and B). Excision of 3′UTR resulted in marked reduction of *Tg*PKG isoforms in the mutant. Densitometry of immunoblots revealed a similar decline in the two isoforms of the mutant clones (135-kD isoform, ~70%; 112-kD isoform, ~85%; Fig 6B).

Cre recombinase–mediated down-regulation of *Tg*PKG protein led to an analogous inhibition of the parasite growth in plaque assays (Fig 6C). Yet again, comparable with the *Tg*ATPase$_P$-GC mutant (Fig 5A), cell division of the *Tg*PKG mutant was only moderately affected, as estimated by a smaller fraction of bigger vacuoles containing 32 or 64 parasites in early (24 h) and late (40 h) cultures (Fig 6D). We scored a noteworthy invasion defect in the *Tg*PKG mutant (Fig 6E). In accord, the mutant exhibited a defective egress at all tested time points (Fig 6F). The motile fraction dropped by almost 50% in the mutant, and trail lengths were accordingly shorter (~24 *μm*) compared with the control strains (~55 *μm*) (Fig 6G). Not least, treatment with C2 further reduced the motile fraction and trail lengths of the mutant and its progenitor (Fig S8). The impact of C2 was somewhat stronger in the mutant, but none of the two strains exhibited a complete inhibition, again resonating with the *Tg*ATPase$_P$-GC mutant (Fig 5E). Collectively, our results clearly show that individual repression of *Tg*ATPase$_P$-GC and *Tg*PKG using a common approach imposes nearly identical phenotypic defects on the lytic cycle.

## Discussion

This study characterized an alveolate-specific protein, termed *Tg*ATPase$_P$-GC herein, which imparts a central piece of cGMP signaling conundrum in *T. gondii*. Our research in conjunction with three recently published independent studies (Brown & Sibley, 2018; Bisio et al, 2019; Yang et al, 2019) provides a comprehensive functional and structural insight into the initiation of cGMP signaling in *T. gondii*. The parasite encodes an unusual and multifunctional protein (*Tg*ATPase$_P$-GC) primarily localized in the plasma membrane at the apical pole of the tachyzoite stage. Previous reports have concluded a surface localization of *Tg*ATPase$_P$-GC by inference, which we reveal to be the plasma

membrane as opposed to the IMC (Fig 2C). Interestingly, a confined localization of GCβ in a unique spot of the ookinete membrane was recently reported to be critical for the protein function in *P. yoelii* (Gao et al, 2018). Similar work in *T. gondii* (Brown & Sibley, 2018) demonstrated that deletion or mutation of ATPase domain in *Tg*ATPase$_P$-GC mislocalized the protein to the ER and cytosol, whereas deletion or mutation of GC domains did not affect the apical localization. An impaired secretion of micronemes was observed in both cases, suggesting the importance of ATPase domain for localization and function. Earlier, the same group reported that the long isoform of *Tg*PKG (*Tg*PKG$^I$) associated with the plasma membrane is essential and sufficient for PKG-dependent events; however, the shorter cytosolic isoform (*Tg*PKG$^{II}$) is inadequate and dispensable (Brown et al, 2017). Our results illustrate that the C terminus of *Tg*ATPase$_P$-GC faces inside the plasmalemma bilayer (Fig 2B), where it should be in spatial proximity with *Tg*PKG$^I$ to allow efficient induction of cGMP signaling. Moreover, we show that *Tg*ATPase$_P$-GC is expressed throughout the lytic cycle of tachyzoites but needed only for their entry or exit from host cells, which implies a post-translational activation of cGMP signaling.

This work along with others (Brown & Sibley, 2018; Bisio et al, 2019; Yang et al, 2019) reveals that *Tg*ATPase$_P$-GC is essential for a successful lytic cycle. Its knockdown by 3′UTR excision using Cre/loxP method demonstrated a physiological role of cGMP for the invasion and egress. In further work, we uncovered that *Tg*PKG depletion phenocopies the *Tg*ATPase$_P$-GC knockdown mutant, resonating with previous work (Brown et al, 2017). In addition, our in silico analysis offers valuable insights into catalytic functioning of GC domains in *Tg*ATPase$_P$-GC. GC1 and GC2 dimerize to form only one pseudo-symmetric catalytic center in contrast to the homodimer formation in mammalian pGCs (Linder & Schultz, 2003; Linder, 2005; Steegborn, 2014). GC1 and GC2 have probably evolved by gene duplication, causing degeneration of the unused second regulatory binding site, as reported in tmACs (Tesmer et al, 1997; Linder, 2005; Steegborn, 2014). The function of *Tg*ATPase$_P$-GC as a guanylate cyclase aligns rather well with its predicted substrate specificity for GTP, although its contribution for cAMP synthesis (if any) remains to be tested. Our attempts to functionally complement an adenylate cyclase mutant of *E. coli* with GC domains or to get catalytically active recombinant proteins from *E. coli* were not fruitful; nonetheless, we could demonstrate that repression of *Tg*ATPase$_P$-GC leads to a comparable reduction in the cGMP level, indicating its function as a guanylate cyclase. Our predicted topology of *Tg*ATPase$_P$-GC harboring 22 transmembrane helices differs from the work of Yang et al (2019), suggesting the occurrence of 19 helices, but echoes with the two other reports (Brown & Sibley, 2018; Bisio et al, 2019).

Unlike the C-terminal GC of *Tg*ATPase$_P$-GC, the function of N-terminal ATPase domain remains rather enigmatic. The latter resembles P4-ATPases similar to other alveolate GCs (Linder et al, 1999; Carucci et al, 2000; Kenthirapalan et al, 2016; Baker et al, 2017). Lipids and cation homeostasis have been shown to influence the gliding motility and associated protein secretion, which in turn drives the egress and invasion events (Endo & Yagita, 1990; Rohloff et al, 2011; Brochet et al, 2014; Bullen et al, 2016; Frénal et al, 2017). There is little evidence however, how lipid and cation-dependent pathways embrace each other. It is thus tempting to propose a

nodal role of $Tg$ATPase$_P$-GC in asymmetric distribution of phospholipids between the membrane leaflets, and cation flux (e.g., $Ca^{2+}$, $K^+$, and $Na^+$) across the plasma membrane. The recent studies (Brown & Sibley, 2018; Bisio et al, 2019) also entail a regulatory role of P4-ATPase domain on the functioning of GC domain. Conversely, the GC domain of GC$\beta$ was found sufficient to produce cGMP independently of the ATPase domain in *P. yoelii* (Gao et al, 2018). This may be due to degenerated conserved sequences in the ATPase domain of $Pf$GC$\beta$, as shown in the sequence alignment (Fig S2). Equally, the expression of $Pf$GC$\alpha$ and $Pf$GC$\beta$ resulted in the functional protein only for $Pf$GC$\beta$, but not for $Pf$GC$\alpha$ (Carucci et al, 2000). Indeed, $Tg$ATPase$_P$-GC is more homologous to $Pf$GC$\alpha$ (identity, 43%; E value, $3E^{-140}$), which may explain the differences in the $Pf$GC$\beta$ and $Tg$ATPase$_P$-GC mutants.

Two additional components, CDC50.1 and UGO, were suggested to secure the functionality of $Tg$ATPase$_P$-GC by interacting with ATPase and GC domains, respectively (Bisio et al, 2019). It was already known that most of the mammalian P4-ATPases require CDC50 proteins as accessory subunits, which are transmembrane glycoproteins ensuring formation of active P4-ATPase complex (Coleman & Molday, 2011; Andersen et al, 2016). A similar interaction between GC$\beta$ and CDC50A protein was shown in *P. yoelii* (Gao et al, 2018). The CDC50.1 expressed in *T. gondii* was also found to bind with the P4-ATPase domain of $Tg$ATPase$_P$-GC to facilitate the recognition of phosphatidic acid and thereby regulate the activation of GC domain. The second interacting partner UGO on the other hand was proposed to be essential for the activation of GC domain after phosphatidic acid binding (Bisio et al, 2019). The study of Yang et al (2019) showed that the depletion of $Tg$ATPase$_P$-GC impairs the production of phosphatidic acid, which is consistent with our postulated lipid flipping function of P4-ATPase domain. However, a systematic experimental analysis is still required to understand the function of P4-ATPase and its intramolecular coordination with GC domains in $Tg$ATPase$_P$-GC.

The topology, subcellular localization, modeled structure, and depicted multifunctionality of $Tg$ATPase$_P$-GC strikingly differ from those of the particulate GCs from mammals. Other distinguished features of mammalian pGCs, such as extracellular ligand binding and regulatory kinase-homology domains, are also absent in $Tg$ATPase$_P$-GC, adding to evolutionary specialization of cGMP signaling in *T. gondii*. Likewise, other PKG-independent effectors of cGMP, that is, nucleotide-gated ion channels as reported in mammalian cells (MacFarland, 1995; Lucas et al, 2000; Pilz & Casteel, 2003), could not be identified in the genome of *T. gondii*, suggesting a rather linear transduction of cGMP signaling through PKG. Notably, the topology of $Tg$ATPase$_P$-GC is shared by members of another alveolate phylum Ciliophora (e.g., *Paramecium* and *Tetrahymena*) (Linder et al, 1999), which exhibit an entirely different lifestyle. Moreover, a similar protein with two GC domains but lacking ATPase-like region is present in *Dictyostelium* (member of amoebozoa) (Roelofs et al, 2001). Such a conservation of cGMP signaling architecture in several alveolates with otherwise diverse lifestyles signifies a convoluted functional repurposing of signaling within the protozoan kingdom. Not least, a divergent origin and essential requirement of cGMP cascade can be exploited to selectively inhibit the asexual reproduction of the parasitic protists.

# Materials and Methods

## Reagents and resources

The culture media and additives were purchased from PAN Biotech. Other common laboratory chemicals were supplied by Sigma-Aldrich. DNA-modifying enzymes were obtained from New England Biolabs. Commercial kits, purchased from Life Technologies and Analytik Jena, were used for cloning into plasmids and for isolation of nucleic acids. Gene cloning and vector amplifications were performed in *E. coli* (XL1B). Oligonucleotides and fluorophore-conjugated secondary antibodies (Alexa488 and Alexa594) were obtained from Thermo Fisher Scientific. The primary antibody ($\alpha$-HA) and zaprinast were acquired from Sigma-Aldrich. PDE inhibitor BIPPO (5-benzyl-3-isopropyl-1H-pyrazolo [4,3$d$] pyrimidin-7($6H$)-one) (Howard et al, 2015) and PKG inhibitor compound 2 (4-[7-[(dimethylamino) methyl]-2-(4-fluorophenyl) imidazo [1,2-$\alpha$] pyridin-3-yl] pyrimidin-2-amine) (Biftu et al, 2005) were donated by Philip Thompson (Monash University, Australia) and Oliver Billker (Wellcome Trust Sanger Institute, UK), respectively. The primary antibodies against $Tg$Gap45 (Plattner et al, 2008), $Tg$Sag1 (Dubremetz et al, 1993), and $Tg$ISP1 (Beck et al, 2010) were provided by Dominique Soldati-Favre (University of Geneva, Switzerland), Jean-François Dubremetz (University of Montpellier, France), and Peter Bradley (University of California, Los Angeles, US), respectively. The RH$\Delta ku80$-$hxgprt^-$ strain of *T. gondii*, lacking nonhomologous end-joining repair to facilitate homologous recombination-mediated genome manipulation (Fox et al, 2009; Huynh & Carruthers, 2009), was offered by Vern Carruthers (University of Michigan, USA). Human foreskin fibroblast (HFF) cells were obtained from Carsten Lüder (Georg-August University, Göttingen, Germany).

## Parasite and host-cell cultures

Tachyzoites of *T. gondii* (RH$\Delta ku80$-$hxgprt^-$ and its derivative strains) were serially maintained by infecting confluent monolayers of HFFs every second day, as described previously (Gupta et al, 2005). Briefly, the host cells were harvested by trypsinization (0.25% trypsin–EDTA) and grown to confluence in flasks, plates or dishes, as required for the experiments. Uninfected and infected HFF cells were cultivated in DMEM with glucose (4.5 g/l) supplemented with 10% heat-inactivated FBS (iFBS; PAN Biotech), 2 mM glutamine, 1 mM sodium pyruvate, 1× minimum Eagle's medium nonessential amino acids (100 $\mu M$ each of serine, glycine, alanine, asparagine, aspartic acid, glutamate, and proline), penicillin (100 U/ml), and streptomycin (100 $\mu$g/ml) in a humidified incubator (37°C, 5% $CO_2$). Freezer stocks of intracellular tachyzoites (HFFs with mature parasite vacuoles) were made in a medium containing 45% inactivated FBS, 5% DMSO, and 50% DMEM with aforementioned supplements.

## Expression of GC1 and GC2 of $Tg$ATPase$_P$-GC in *E. coli*

Heterologous expression of the GC1 and GC2 domains was performed in the M15 and BTH101 strains of *E. coli*, for protein purification and functional complementation, respectively (see results

below). The open reading frames of GC1 (2,850–3,244 bp), GC2 (3,934–4,242 bp), and GC1+GC2 (2,850–4,242 bp) domains starting with the upstream start codon (ATG) were amplified from the tachyzoite mRNA (RHΔ*ku80-hxgprt*⁻). The first-strand cDNA used for ORF-specific PCR was generated from the total RNA by oligo-T primers using a commercial kit (Life Technologies). The ORFs were cloned into the *pQE60* vector at the *Bgl*II restriction site, resulting in a C-terminal 6xHis-tag (primers in Table S1). To express and purify the indicated proteins, 5 ml culture of the recombinant M15 strains (grown overnight at 37°C) were diluted to an $OD_{600}$ of 0.1 in 100 ml of Luria–Bertani (LB) medium containing 100 $\mu$g/ml ampicillin and 50 $\mu$g/ml kanamycin and incubated at 37°C until $OD_{600}$ reached to 0.4–0.6 (4–5 h). The cultures were then induced with 0.1 mM IPTG overnight at 25°C.

The bacterial cell lysates were prepared under denaturing conditions, and proteins were purified using Ni-NTA column according to the manufacturer's protocol (Novex by Life Technologies). Briefly, the cells were harvested (3,000*g*, 20 min, 4°C) and resuspended in 8 ml lysis buffer (6 M guanidine HCl, 20 mM $NaH_2PO_4$, and 500 mM NaCl, pH 7.8) by shaking for 10 min. They were disrupted by probe sonication (5 pulses, 30 s each with intermittent cooling on ice-cold water) and flash-frozen in liquid nitrogen followed by thawing at 37°C (3×). Intact cells were removed by pelleting at 3,000*g* for 15 min. Lysate-containing supernatant (cell-free extract) was loaded on an Ni-NTA column pretreated with binding buffer (8 M urea, 20 mM $NaH_2PO_4$, and 500 mM NaCl, pH 7.8) ensued by two washings with 4 ml washing buffer (8 M urea, 20 mM $NaH_2PO_4$, and 500 mM NaCl) with pH 6.0 and pH 5.3, respectively. Proteins were eluted in 5 ml of elution buffer (20 mM $NaH_2PO_4$, 100 mM NaCl, and 10% glycerol, pH 7.8). The eluate was concentrated and dialyzed by centrifugal filters (30-kD cutoff, Amicon ultra filters; Merck Millipore). A refolding of the purified proteins led to precipitation. The amount of urea was, thus, gradually reduced to 0.32 M by adding 4× volume of the buffer with 20 mM $NaH_2PO_4$, 100 mM NaCl, and 10% glycerol in successive centrifugation steps. The final protein preparation was stored at –80°C in liquid nitrogen.

The function of GC1, GC2, and GC1+GC2 as being potential adenylate cyclase domains was tested in the *E. coli* BTH101 strain, which lacks cAMP signaling because of enzymatic deficiency and thus unable to use maltose as a carbon resource (Karimova et al, 1998). The *pQE60* constructs encoding indicated ORF sequences were transformed into the BTH101 strain. The bacterial cultures were grown overnight in 5 ml of LB medium containing 100 $\mu$g/ml ampicillin and 100 $\mu$g/ml streptomycin at 37°C. Protein expression was induced by 200 $\mu$M IPTG (2 h, 30°C) followed by serial dilution plating on MacConkey agar (pH 7.5) supplemented with 1% maltose, 200 $\mu$M IPTG, and 100 $\mu$g/ml of each antibiotic. The strain harboring the *pQE60* vector served as a negative control, whereas the plasmid expressing *Cya*A (native bacterial adenylate cyclase) was included as a positive control. Agar plates were incubated at 30°C (~32 h) to examine for the appearance of colonies.

## GC assay

GC1 and GC2 (3 $\mu$g) purified from M15 strain of *E. coli* were examined by GC assay. The enzymatic reaction (100 $\mu$l) was executed in 50 mM Hepes buffer (pH 7.5) containing 100 mM NaCl, 2 mM $MnCl_2$, and 2 mM GTP (22°C, 650 rpm, 10 min). The assay was quenched by adding 200 $\mu$l of 0.1 M HCl, followed by high-performance liquid chromatography or cGMP-specific ELISA. GTP and cGMP (2 mM each) were used as standards for HPLC.

## Genomic tagging of *Tg*ATPase$_P$-GC and *Tg*PKG

Sequences of *Tg*ATPase$_P$-GC (TGGT1_254370) and *Tg*PKG (TGGT1_311360) genes were obtained from the parasite genome database (ToxoDB) (Gajria et al, 2008). The expression and subcellular localization of *Tg*ATPase$_P$-GC and *Tg*PKG were determined by 3′-insertional tagging (3′IT) of corresponding genes with an epitope, essentially as reported before (Nitzsche et al, 2017). To achieve this, the 3′ end of the gene (1–1.5 kb 3′-crossover sequence or 3′COS) was amplified from gDNA of the RHΔ*ku80-hxgprt*⁻ strain using Q5 High-Fidelity DNA Polymerase (Bio-Rad Laboratories) (see Table S1 for primers). The HA-tagged amplicons were cloned into the *p3′IT-HXGPRT* vector using *Xcm*I/*Eco*RI and *Nco*I/*Eco*RI enzyme pairs for *Tg*ATPase$_P$-GC and *Tg*PKG, respectively. This vector harbors 3′UTR of the *Tg*Gra1 gene (to stabilize the transcript) as well as a drug selection cassette expressing HXGPRT, as indicated (Figs 2A and S7A). The constructs were linearized (15 $\mu$g) in the first half of COS using *Sac*I (*Tg*ATPase$_P$-GC) and *Sph*I (*Tg*PKG), followed by transfection into tachyzoites of the RHΔ*ku80-hxgprt*⁻ strain ($10^7$). In this regard, extracellular parasites were pelleted (420*g*, 10 min), and suspended in filter-sterile cytomix (120 mM KCl, 0.15 mM $CaCl_2$, 10 mM $K_2HPO_4$/$KH_2PO_4$, 25 mM Hepes, 2 mM EGTA, and 5 mM $MgCl_2$, pH 7.4) supplemented with fresh ATP (2 $\mu$M) and glutathione (5 $\mu$M). Tachyzoites were transfected by electroporation using the Amaxa electroporator (T-016 program; voltage, 1,700 V; resistance, 50 Ω; pulse duration, 176 $\mu$s; number of pulses, 2; interval, 100 ms; polarity, and unipolar). Transgenic parasites expressing HXGPRT were selected with mycophenolic acid (25 $\mu$g/ml) and xanthine (50 $\mu$g/ml) (Donald et al, 1996).

A successful epitope-tagging of the genes was verified by recombination-specific PCR and sequencing of subsequent amplicons. The stable drug-resistant transgenic parasites were subjected to limiting dilution in 96-well plates with confluent HFF cells to obtain the clonal lines for downstream analyses. The eventual strains expressed *Tg*ATPase$_P$-GC-HA$_{3′IT}$ or *Tg*PKG-HA$_{3′IT}$ under the control of their native promoter and *Tg*Gra1-3′UTR. Using a similar strategy, we generated additional transgenic strains, in which *Tg*Gra1-3′UTR was replaced by the native 3′UTR of *Tg*ATPase$_P$-GC and *Tg*PKG genes. Here, nearly 1 kb of 3′UTR beginning from the translation stop codon of the *Tg*ATPase$_P$-GC and *Tg*PKG genes was amplified from gDNA of the RHΔ*ku80-hxgprt*⁻ strain and then cloned into the *p3′IT-HXGPRT* plasmid (harboring 3′COS of individual genes) at *Eco*RI/*Spe*I sites, substituting for *Tg*Gra1-3′UTR (primers in Table S1). The constructs were linearized and transfected into parasites, followed by drug selection, crossover-specific PCR screening, and limiting dilution to obtain the clonal transgenic strains, as described above.

## CRISPR/Cas9-assisted genetic disruption of *Tg*ATPase$_P$-GC

The mutagenesis of *Tg*ATPase$_P$-GC was achieved using CRISPR/Cas9 system, as reported previously for other genes (Sidik et al, 2014). To

express gene-specific sgRNA and Cas9, we used *pU6-sgRNA-Cas9* vector. The oligonucleotide pair, designed to target the nucleotide region from 145 to 164, was cloned into vector by golden gate assembly method using *Tg*ATPase$_P$-GC-sgRNA-F1/R1 primers (Table S1). The assembly was initiated by mixing the *pU6-sgRNA-Cas9* vector (45 ng), *Bsa*I-HF enzyme (5 U; New England Biolabs), oligonucleotides (0.5 µM), T4 ligase (5 U; ThermoFisher Scientific) in a total volume of 20 µl. The conditions were set as 37°C (2 min) for *Bsa*I digestion and 20°C (5 min) for ligation, and repeated for 30 cycles, followed by incubation at 37°C (10 min) before T4 ligase inactivation at 50°C (10 min) and *Bsa*I inactivation at 80°C (10 min). The product was directly transformed into XL1B strain of *E. coli*. Positive clones were verified by DNA sequencing, followed by transfection of 15 µg construct into the P$_{native}$-*Tg*ATPase$_P$-GC-HA$_{3'IT}$-*Tg*Gra1-3′UTR strain (stated as the progenitor strain in Fig 3A) to disrupt the gene. A Cas9-mediated cleavage at the *Tg*ATPase$_P$-GC locus caused a loss of HA signal in transfected parasites, which was monitored by IFAs at various time points of cultures (Fig 3A).

## Cre-mediated knockdown of *Tg*ATPase$_P$-GC and *Tg*PKG by excision of 3′UTR

A knockdown of the genes of interest was performed by Cre recombinase–mediated excision of the loxP-flanked (floxed) 3′UTR in the HA-tagged strains. The progenitor strains (P$_{native}$-*Tg*ATPase$_P$-GC-HA$_{3'IT}$-3′UTR$_{floxed}$ or P$_{native}$-*Tg*PKG-HA$_{3'IT}$-3′UTR$_{floxed}$) were transfected with a plasmid (*pSag1-Cre*) expressing Cre recombinase that recognizes and excises the loxP sites flanking 3′UTR and HXGPRT selection cassette (Figs 4A and S7A). Tachyzoites transfected with Cre-encoding vector were then negatively selected for the loss of HXGPRT expression using 6-thioxanthine (80 µg/ml) (Donald et al, 1996). The single clones with Cre-excised 3′UTR were screened by PCR using indicated primers (Table S1) and validated by sequencing. The expression level of the target proteins in the mutants was confirmed by immunofluorescence and immunoblot analysis, as described below. For each phenotyping assay, the mutant parasites were generated fresh by transfecting Cre expression plasmid into the progenitor strain followed by drug selection and isolation of clones by PCR screening. The mutants were not propagated beyond 2–3 wk to minimize any adaptation in culture.

## Measurement of cGMP in tachyzoites

Confluent HFF monolayers were infected either with the parental (RHΔ*ku80-hxgprt⁻*, MOI: 1.3), progenitor (P$_{native}$-*Tg*ATPase$_P$-GC-HA$_{3'IT}$-3′UTR$_{floxed}$, MOI: 1.5), or the knockdown mutant (P$_{native}$-*Tg*ATPase$_P$-GC-HA$_{3'IT}$-3′UTR$_{excised}$, MOI: 2) strain for 36–40 h. Infected cells containing mature parasite vacuoles were then washed twice with ice-cold PBS to eliminate free parasites, scraped by adding 2 ml colorless DMEM, and extruded through a 27G syringe (2×), followed by centrifugation (420*g*, 10 min, 4°C). The parasite pellets were dissolved in 100 µl of cold colorless DMEM for counting and cGMP extraction. The parasite suspension (5 × 10$^6$, 100 µl) was mixed with 200 µl of ice-cold 0.1 M HCl, incubated for 20 min at room temperature, and flash-frozen in liquid nitrogen until used. The samples were thawed and squirted through pipette to disrupt the parasite membranes. The

colorless DMEM and HFF cells, treated similarly, were used as negative controls. The samples were transferred onto centrifugal filters (0.22-µm, Corning Costar Spin-X, CLS8169; Sigma-Aldrich) to eliminate the membrane particulates (20,800 g, 10 min, 4°C). The flow-through was filtered once again via 10-kD filter units (Amicon Ultra-0.5 ml filters; Millipore) to obtain pure samples (20,800 g, 30 min, 4°C), which were then subjected to ELISA using the commercial Direct cGMP ELISA kit (ADI-900-014; Enzo Life Sciences) to measure cGMP levels of parasites. The acetylated (2 h) format of the assay was run for all samples, including the standards and controls, as described by the manufacturer. The absorbance was measured at 405 nm; the data were adjusted for the dilution factor (1:3) and analyzed using the microplate analysis tool (www.myassays.com).

## Indirect IFA

IFA with extracellular and intracellular parasites was executed, as described elsewhere (Kong et al, 2018). Freshly egressed tachyzoites were incubated (20 min) on the BSA-coated (0.01%) coverslips to stain the extracellular stage. Intracellular parasites were stained within confluent monolayers of HFFs grown on coverslips (24–32 h postinfection). The samples were fixed with 4% paraformaldehyde (15 min) followed by neutralization with 0.1 M glycine in PBS (5 min). For standard IFA, the samples were permeabilized with 0.2% Triton-X 100/PBS (20 min), and nonspecific binding was blocked by 2% BSA in 0.2% Triton X-100/PBS (20 min). Afterwards, parasites were stained with specified primary antibodies (rabbit/mouse *α*-HA, 1:3,000; rabbit *α*-*Tg*Gap45, 1:8,000; mouse *α*-*Tg*Sag1, 1:10,000; mouse *α*-*Tg*ISP1, 1:2,000) suspended in 2% BSA in 0.2% Triton-X 100/PBS (1 h). The cells were washed 3× with 0.2% Triton X-100/PBS. The samples were incubated with appropriate Alexa 488– and Alexa 594–conjugated secondary antibodies (1 h), followed by 3× washing with PBS. Immunostained cells were mounted with Fluoromount G containing DAPI for nuclei staining, and then imaged with Zeiss Apotome microscope (Zeiss).

To resolve the C-terminal topology of *Tg*ATPase$_P$-GC, fresh extracellular parasites were stained with rabbit *α*-HA (1:3,000) and mouse *α*-*Tg*Sag1 (1:10,000) antibodies, or with rabbit *α*-*Tg*Gap45 (1:8,000) and mouse *α*-*Tg*Sag1 (1:10,000) before and after permeabilization as described elsewhere (Blume et al, 2009). 5 × 10$^4$ parasites were fixed on BSA-coated coverslips using 4% paraformaldehyde with 0.05% glutaraldehyde. Permeabilized cells were subjected to immunostaining as indicated above, except for that all solutions were substituted to PBS for nonpermeabilized staining (i.e., no detergent and BSA). To test the membrane location of *Tg*ATPase$_P$-GC, the IMC was separated from the plasma membrane by treating extracellular parasites with *α*-toxin from *Clostridium septicum* (20 nM, 2 h) (List Biological Laboratories), followed by fixation on BSA-coated coverslips and antibody staining. In both cases, the standard immunostaining procedure was performed subsequently.

## Lytic cycle assays

All assays were set up with fresh syringe-released parasites, essentially the same as reported earlier (Arroyo-Olarte et al, 2015). Parasitized cultures (MOI: 2; 40–44 h post-infection) were washed

with standard culture medium, scraped, and extruded through a 27G syringe (2×). For plaque assays, HFF monolayers grown in six-well plates were infected with tachyzoites (150 parasites per well) and incubated for 7 d without perturbation. The cultures were fixed with ice-cold methanol (–80°C, 10 min) and stained with crystal violet solution (12.5 g dye in 125 ml ethanol mixed with 500 ml 1% ammonium oxalate) for 15 min, followed by washing with PBS. The plaque sizes were measured by using the ImageJ software (NIH). To set up the replication assays, host cells grown on coverslips placed in 24-well plates were infected with $3 \times 10^4$ parasites before fixation, permeabilization, neutralization, blocking, and immunostaining with $\alpha$-$Tg$Gap45 and Alexa594 antibodies, as explained in IFA. The cell division was assessed by enumerating intracellular parasites within their vacuoles. To measure the gliding motility, $4 \times 10^5$ parasites suspended in calcium-free HBSS with or without drugs (BIPPO, 55 $\mu$M; zaprinast, 500 $\mu$M; and compound 2, 2 $\mu$M) were incubated first to let them settle (15 min, room temperature) and glide (15 min, 37°C) on BSA-coated (0.01%) coverslips. The samples were subjected to IFA using $\alpha$-$Tg$Sag1 and Alexa488 antibodies, as mentioned above. Motile fractions were counted on the micro-scope, and trail lengths were quantified using the ImageJ software.

For invasion and egress, host-cell monolayers cultured on glass coverslips were infected with MOI: 10 for 1 h, or with MOI: 1 for 40–64 h, respectively. For invasion, BIPPO (55 $\mu$M), zaprinast (500 $\mu$M), and compound 2 (2 $\mu$M) treatments were performed during the 1-h incubation time; however, the effect of drugs on the parasite egress were tested by incubating for 5 min 30 s (40 h postinfection). The cells were subsequently fixed with 4% paraformaldehyde (15 min), neutralized by 0.1 M glycine/PBS (5 min), and then blocked in 3% BSA/PBS (30 min). Noninvasive parasites or egressed vacuoles were stained with $\alpha$-TgSag1 antibody (mouse, 1:10,000, 1 h) before de-tergent permeabilization. The cultures were washed 3× with PBS, permeabilized with 0.2% Triton-X 100/PBS (20 min), and stained with $\alpha$-TgGap45 antibody (rabbit, 1:8,000, 1 h) to visualize intracellular parasites. The samples were then washed and immunostained with Alexa 488– and Alexa 594–conjugated antibodies (1:10,000, 1 h). The fraction of invaded parasites was counted by immunostaining with $\alpha$-TgGap45/Alexa594 (red) but not with $\alpha$-TgSag1/Alexa488 (green). The percentage of lysed vacuole was scored directly by counting $\alpha$-$Tg$Sag1/Alexa488 (green) and dual-colored parasites.

## Immunoblot analysis

Standard Western blot was performed to determine the expression level of $Tg$PKG-HA$_{3'IT}$, whereas the dot blot analysis was undertaken for $Tg$ATPase$_P$-GC-HA$_{3'IT}$ because of its large size (477-kD). For the former assay, the protein samples prepared from extracellular parasites ($2 \times 10^7$) were separated by 8% SDS–PAGE (120 V) followed by semidry blotting onto a nitrocellulose membrane (85 mA/cm$^2$, 3 h). The membrane was blocked with 5% skimmed milk solution prepared in 0.2% Tween 20/TBS (1 h with shaking at room tem-perature), and then stained with rabbit $\alpha$-HA (1:1,000) and mouse $\alpha$-$Tg$Rop2 (1:1,000) antibodies. For the dot blot, protein samples equivalent to $10^7$ parasites were spotted directly onto nitrocellulose membrane. The membrane was blocked in a solution containing 1% BSA and 0.05% Tween 20 in TBS for 1 h, followed by immunostaining with rabbit $\alpha$-HA (1:1,000) and/or rabbit $\alpha$-$Tg$Gap45 (1:3,000)

antibodies diluted in the same buffer. Proteins were visualized by Li-COR imaging after staining with IRDye 680RD and IRDye 800CW (1:15,000) antibodies. Densitometric analysis was performed using the ImageJ software, as reported elsewhere (https://imagej.nih.gov/ij/docs/examples/dot-blot).

## Structure modeling

The membrane topology of $Tg$ATPase$_P$-GC was assessed based on the data obtained from TMHMM (Sonnhammer et al, 1998), SMART (Letunic et al, 2014), Phobius (Käll et al, 2004), NCBI-conserved domain search (Marchler-Bauer & Bryant, 2004), and TMpred (Hofmann & Stoffel, 1993) (Fig 1A). To detect conserved residues of the active sites in $Tg$ATPase$_P$-GC, GC1 and GC2 regions were aligned with the cyclase domains of representative organisms using Clustal Omega program (Sievers et al, 2011). Similarly, conserved motifs in the ATPase domain of $Tg$ATPase$_P$-GC were obtained by alignment with members of human P4-ATPases using MAFFT online alignment server (v7) (Katoh & Standley, 2013). Conserved residues were color-coded by the Clustal Omega program.

The conserved motifs and cyclase domains for the tertiary model were predicted using UniProt (https://www.uniprot.org/). The catalytic units of GC1 (aa 2,929–3,200, lacking the loop from aa 3,038 to 3,103) and GC2 (aa 3,989–4,195) were modeled by SWISS-MODEL (https://swissmodel.expasy.org/), based on a ligand-free tmAC as the structural template (UniProt ID: 1AZS). The Qualitative Model Energy Analysis and the Global Model Quality Estimation scores were determined to be –3.44 and 0.59, respectively, reflecting the accuracy of the model. Subsequently, the ligand GTP$\alpha$S was positioned into the model of pseudo-heterodimer corresponding to the location of ATP$\alpha$S in tmAC (Protein Data Bank ID: 1CJK).

## Phylogenetic analysis

The open reading frame sequences of $Tg$ATPase$_P$-GC orthologs were obtained from the NCBI database. Briefly, the whole se-quences of 30 proteins were aligned, and based on this alignment, a consensus tree was generated using the CLC Genomics Workbench v12.0 (QIAGEN Bioinformatics). Maximum likelihood method was used for clustering; bootstrap analysis was performed with 100 iterations; neighborhood joining was used for construction; and JJT model was selected for amino acid substitutions. The eventual tree was visualized as a cladogram by Figtree v1.4.3 (http://tree.bio.ed.ac.uk/software/figtree/), followed by text annotation in the Microsoft PowerPoint.

## Data analyses and statistics

All experiments were performed at least three independent times, unless specified otherwise. Figures illustrating images or making of transgenic strains typically show a representative of three or more biological replicates. Graphs and statistical significance were generated using GraphPad Prism v6.0. The error bars in graphs signify means with SEM from multiple assays, as indicated in figure

legends. The *P*-values were calculated by *t* test (*$*P* ≤ 0.05; **$*P* ≤ 0.01; ***$*P* ≤ 0.001; and ****$*P* ≤ 0.0001).

## Supplementary Information

## Acknowledgements

We thank Grit Meusel (Humboldt University, Berlin) for technical assistance. We are grateful to David Baker (London School of Hygiene and tropical Medicine, UK) for his initial review of this manuscript. In addition, we appreciate the parasitology community for sharing certain reagents used in this study. The work was supported by a research grants (GU1100/7-1) and a Heisenberg program grant (GU1100/13), both awarded to N Gupta by German Research Foundation (DFG). Financial support to Ö Günay-Esiyok was provided through Elsa-Neumann-Scholarship by the state of Berlin, Germany.

### Author Contributions

Ö Günay-Esiyok: data curation, formal analysis, validation, investigation, visualization, methodology, and writing—original draft, review, and editing.
U Scheib: data curation, visualization, and writing—review and editing.
M Noll: resources and writing—review and editing.
N Gupta: conceptualization, resources, data curation, software, formal analysis, supervision, funding acquisition, investigation, methodology, project administration, and writing—original draft, review, and editing.

### Conflict of Interest Statement

The authors declare that they have no conflict of interest.

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
