## [Reviewer comments · Life Science Alliance]

An Unusual and Vital Protein with Guanylate Cyclase and P4-ATPase Domains in a Pathogenic Protist

L'wibj Dr'nay-Esiyok, Ulrike Scheib, Matthias Noll and Nishith Gupta

DOI: 10.26508/lsa.201900402

Review timeline:	Submission Date:	14 April 2019
	Editorial Decision:	13 May 2019
	Revision Received:	5 June 2019
	Editorial Decision:	5 June 2019
	Revision Received:	6 June 2019
	Accepted:	6 June 2019

Report:

(Note: Letters and reports are not edited. The original formatting of letters and referee reports may not be reflected in this compilation.)

1st Editorial Decision

13 May 2019

Thank you for submitting your manuscript entitled "An Unusual and Vital Protein with Guanylate Cyclase and P-ATPase Domains in a Pathogenic Protist" to Life Science Alliance. The manuscript was assessed by expert reviewers, whose comments are appended to this letter.

As you will see, the reviewers support the publication of your work in Life Science Alliance pending satisfactory revision, and I would thus like to invite you to submit a revised version to us. Importantly, recent work should get introduced and discussed more extensively, also including the work by Tonkin and colleagues that got published after you submitted your manuscript to us. Most other comments of the reviewers can also get addressed by text changes, and in the interest of time, specific comments 1 (western blot analysis) and 3 (HA-staining) of reviewer #2 do not mandatorily need to get addressed experimentally. Adding insight into a potential adenylate cyclase role (reviewer #3) would strengthen your manuscript tremendously, so please consider testing this. I envision a three week revision time and I would be happy to discuss the revision requirements further with you.

Thank you for this interesting contribution to Life Science Alliance. We are looking forward to receiving your revised manuscript.

2nd Editorial Decision

5 June 2019

Thank you for submitting your revised manuscript entitled "An Unusual and Vital Protein with Guanylate Cyclase and P4-ATPase Domains in a Pathogenic Protist". I appreciate the introduced changes and would be happy to publish your paper in Life Science Alliance pending final revisions necessary to meet our formatting guidelines:

- please upload the suppl table as a docx file
- please upload the suppl figures as individual figure files
- please move the suppl figure legends into the main manuscript file
- please include the suppl methods and results in the main manuscript file

3rd Editorial Decision

6 June 2019

Thank you for submitting your Research Article entitled "An Unusual and Vital Protein with Guanylate Cyclase and P4-ATPase Domains in a Pathogenic Protist". It is a pleasure to let you know that your manuscript is now accepted for publication in Life Science Alliance. Congratulations on this interesting work.